

# DCMIP2016: A Review of Non-hydrostatic Dynamical Core Design and Intercomparison of Participating Models

Paul A. Ullrich[1], Christiane Jablonowski[2], James Kent[3], Peter H. Lauritzen[4], Ramachandran Nair[4],
Kevin A. Reed[5], Colin M. Zarzycki[4], David M. Hall[6], Don Dazlich[7], Ross Heikes[7], Celal Konor[7],
David Randall[7], Thomas Dubos[8], Yann Meurdesoif[8], Xi Chen[9], Lucas Harris[9], Christian Kühnlein[10],
Vivian Lee[11], Abdessamad Qaddouri[11], Claude Girard[11], Marco Giorgetta[12], Daniel Reinert[13],
Joseph Klemp[4], Sang-Hun Park[4], William Skamarock[4], Hiroaki Miura[14], Tomoki Ohno[14],
Ryuji Yoshida[15], Robert Walko[16], Alex Reinecke[17], and Kevin Viner[17]

[1]University of California, Davis
[2]University of Michigan
[3]University of South Wales
[4]National Center for Atmospheric Research
[5]Stony Brook University
[6]University of Colorado, Boulder
[7]Colorado State University
[8]Institut Pierre-Simon Laplace (IPSL)
[9]Geophysical Fluid Dynamics Laboratory (GFDL)
[10]European Centre for Medium-Range Weather Forecasts (ECMWF)
[11]Environment and Climate Change Canada
[12]Max Planck Institute for Meteorology
[13]Deutscher Wetterdienst (DWD)
[14]University of Tokyo
[15]RIKEN
[16]University of Miami
[17]Naval Research Laboratory

*Correspondence to:* Paul A. Ullrich (paullrich@ucdavis.edu)

**Abstract.** Atmospheric dynamical cores are a fundamental component of global atmospheric modeling systems, and are responsible for capturing the dynamical behavior of the Earth's atmosphere via numerical integration of the Naviér-Stokes equations. These systems have existed in one form or another for over half of a century, with the earliest discretizations having now evolved into a complex ecosystem of algorithms and computational strategies. In essence, no two dynamical cores are alike, and their individual successes suggest that no perfect model exists. To better understand modern dynamical cores, this paper aims to provide a comprehensive review of eleven dynamical cores, drawn from modeling centers and groups that participated in the 2016 Dynamical Core Model Intercomparison Project (DCMIP) workshop and summer school. This review includes choice of model grid, variable placement, vertical coordinate, prognostic equations, temporal discretization, and the diffusion, stabilization, filters and fixers employed by each system.



# 1 Introduction

The Dynamical Core Model Intercomparison Project (DCMIP) is an ongoing effort targeting the intercomparison of a fundamental component of global atmospheric modeling systems: the dynamical core. Although this component's role is simply to solve the equations of fluid motion governing atmospheric dynamics (the Naviér-Stokes equations), there are numerous confounding factors and compromises that arise from making global simulations computationally feasible. These factors include the choice of model grid, variable placement, vertical coordinate, prognostic equations, representation of topography, numerical method, temporal discretization, physics/dynamics coupling frequency, and the manner in which artificial diffusion, stabilization, filters and/or energy/mass fixers are applied.

To advance the intercomparison project and provide a unique educational opportunity for students, DCMIP hosted a multidisciplinary two-week summer school and model intercomparison project, held at the National Center for Atmospheric Research (NCAR) in June 2016, that invited graduate students, postdocs, atmospheric modelers, expert lecturers and computer specialists to create a stimulating, unique and hands-on driven learning environment. The 2016 workshop and summer school followed from earlier DCMIP and dynamical core workshops (held in 2012 and 2008, respectively), and other model intercomparison efforts. Its goals were to provide an international forum for discussing outstanding issues in global atmospheric models, and provide a unique training experience for the future generation of climate scientists. Special attention was paid to the role of simplified physical parameterizations, physics-dynamics coupling, non-hydrostatic atmospheric modeling, and variable-resolution global modeling. The summer school and model intercomparison project promoted active learning, innovation, discovery, mentorship and the integration of science and education. Modeling groups were then invited to contribute model descriptions and results to the intercomparison effort for publication.

The summer school directly benefited its participants by providing a unique educational experience and an opportunity to interact with modeling teams from around the world. The workshop is expected to have further repercussions on the development of operational atmospheric modeling systems, by allowing modeling groups to assess their models in the context of the global dynamical core ecosystem. Past and present intercomparison efforts have been leveraged by modeling groups to improve their own models, in turn leading to a positive impact on the quality of weather and climate simulations. The workshop component of DCMIP has also advanced our knowledge of (1) the relative behaviors exhibited by atmospheric dynamical cores, (2) differences that arise among mechanisms for coupling the physical parameterizations and dynamical core, and (3) the impacts of variable-resolution refinement regions and transition zones in global atmospheric simulations. Notably, the use of idealized test cases to isolate specific phenomena gave us a unique opportunity to assess specific differences that arise due to the choice of dynamical core. Another important outcome of the workshop was the development of a standard test case suite and benchmark set of simulations that can be used for assessment of any future dynamical core. The test cases introduced in the 2016 workshop build on the previous DCMIP test case suites (Jablonowski et al., 2008; Ullrich et al., 2012) with tests that now incorporate simplified moist physics.

This paper is the first in a series of papers documenting the results of this workshop. Its purpose is twofold: First, to review the multitude of technologies and techniques that have been developed for non-hydrostatic global atmospheric modeling; and



second, to provide a mechanism to understand the differences that arise in the test cases of later papers in this series. For ease of reference, a list of mathematical symbols that are employed in this paper (and subsequent DCMIP papers) is given in Table 1. Section 2 then provides a brief overview of each of the participating models, along with a tabulation of relevant details about the dynamical core design. The body of this paper is dedicated to an overview of techniques available for building

the infrastructure of a global dynamical core: section 3 describes aspects of the horizontal discretization, including model grids and horizontal placement of prognostic variables; section 4 describes the vertical placement of model variables and choice of vertical coordinates; section 5 describes aspects of variable placement and prognosis; section 6 describes diffusion, stabilization, filters and fixers employed by these models; and section 7 describes temporal discretizations. The summary and conclusions then follow in section 8. Finally, appendix A provides a comprehensive overview of the various forms the

Naviér-Stokes equations take in dynamical cores, and has been included as a resource for dynamical core developers.

## 2  Dynamical cores

This section provide a brief overview of key discretization choices, along with unique features or design specifications from participating dynamical cores. Further details on these choices can be found in subsequent sections. In total, simulation results and model descriptions have been submitted from eleven dynamical cores (see Table 2). The prognostic variables employed

and horizontal discretizations for these dynamical cores are summarized in Table 3. The vertical staggering of variables and vertical coordinate choice is summarized in Table 4. Standard options for diffusion, stabilization, filters, or fixers along with the temporal discretization for these models is summarized in Table 5. A brief description of each participant model follows, focused on the unique features and decisions underlying the model design.

### 2.1  Accelerated Climate Model for Energy–Atmosphere (ACME–A)

20 The ACME–A model has much in common with the Community Atmosphere Spectral Element Model (CAM-SE) (Dennis et al., 2012) as both share a common origin in the High Order Method Modeling Environment (HOMME) (Taylor and Fournier, 2010). ACME-A employs both a hydrostatic model and an experimental non-hydrostatic compressible shallow-atmosphere model. Both variants are designed to be mass and energy conserving, with nearly optimal parallel scalability at large core counts. ACME-A partitions the globe horizontally via an unstructured grid of quadrilateral elements arranged in a cubed-

25 sphere configuration, although unstructured regionally-refined meshes with conforming edges may also be employed. The fluid equations are discretized using dimensional splitting, with a nodal 4th-order spectral element discretization in the horizontal and a vertical floating Lagrangian levels in hybrid terrain-following pressure coordinates $\eta$. Vertical operators are based on the mimetic (mass and energy conserving) 2nd order finite difference discretization of Simmons and Burridge (1981). All fields are co-located in the horizontal, in the sense that they share the same 4th-order basis functions.





**Table 1.** A standard list of symbols used throughout this paper and in the DCMIP.

| Symbol | Description |
|---|---|
| $\lambda$ | Longitude (in radians) |
| $\varphi$ | Latitude (in radians) |
| $z$ | Height with respect to mean sea level (set to zero) |
| $s$ | Vertical model coordinate |
| $p_s$ | Surface pressure ($p_s$ of moist air if $q > 0$) |
| $\Phi_s$ | Surface geopotential |
| $z_s$ | Surface elevation with respect to mean sea level (set to zero) |
| $u$ | Zonal wind velocity |
| $v$ | Meridional wind velocity |
| $w$ | Vertical wind velocity |
| $\mathbf{u}$ | 3D wind vector |
| $\mathbf{u}_h$ | Horizontal wind vector |
| $\mathbf{v}_h$ | Horizontal wind vector with covariant components |
| $\omega$ | Vertical pressure velocity |
| $D$ | Divergence of the horizontal wind vector |
| $\zeta$ | Vertical component of relative vorticity |
| $p$ | Pressure (pressure of moist air if $q > 0$) |
| $e$ | Internal energy |
| $\rho$ | Total air density |
| $\rho_d$ | Dry air density |
| $\rho_s$ | Pseudo-density |
| $T$ | Temperature |
| $T_v$ | Virtual temperature |
| $\theta$ | Potential temperature |
| $\theta_v$ | Virtual potential temperature |
| $\theta_{il}$ | Ice-liquid potential temperature |
| $\theta_\rho$ | Density potential temperature |
| $q$ | Specific humidity |
| $q_v$ | Water vapor mixing ratio |
| $q_c$ | Cloud water mixing ratio |
| $q_r$ | Rain water mixing ratio |
| $q_i$ | General tracer mixing ratio |



**Table 2.** Participating modeling centers and associated dynamical cores that have submitted a model description and/or simulation results.

| Short Name | Long Name | Modeling Center or Group |
|---|---|---|
| ACME–A | Atmosphere model of the Accelerated Climate Model for Energy | Sandia National Laboratories and University of Colorado, Boulder, USA |
| CSU | Colorado State University Model | Colorado State University, USA |
| DYNAMICO | DYNAMICO | Institut Pierre Simon Laplace (IPSL), France |
| FV$^3$ | GFDL Finite-Volume Cubed-Sphere Dynamical Core | Geophysical Fluid Dynamics Laboratory, USA |
| FVM | Finite Volume Module of the Integrated Forecasting System | European Centre for Medium-Range Weather Forecasts |
| GEM | Global Environmental Multiscale model | Environment and Climate Change Canada |
| ICON | Icosahedral Non-hydrostatic model | Max-Planck-Institut für Meteorologie, Germany |
| MPAS | Model for Prediction Across Scales | National Center for Atmospheric Research, USA |
| NICAM | Non-hydrostatic Icosahedral Atmospheric Model | AORI / JAMSTEC / AICS, Japan |
| OLAM | Ocean Land Atmosphere Model | Duke University / University of Miami, USA |
| TEMPEST | Tempest Non-hydrostatic Atmospheric Model | University of California, Davis, USA |

**Table 3.** Details on the prognostic variables and horizontal discretization for participating dynamical cores. Equation set indicates whether a model is hydrostatic (H) or non-hydrostatic (NH), and whether the model presently supports the deep-atmosphere formulation (D). Only three numerical methods are represented among participating models, namely finite-difference (FD), finite-volume (FV), and spectral-element (SE). More details on horizontal staggering can be found in section 3.8.

| Short Name | Equation Set | Prognostic Variables | Horizontal Grid | Numerical method | Horizontal staggering |
|---|---|---|---|---|---|
| ACME–A | H/NH | $\mathbf{u}_h, w, \rho_s, \rho_s\theta, \Phi, \rho_s q_i$ | Cubed-sphere (§3.2) | SE | A-grid |
| CSU | NH (Unified) | $\zeta, D, w, p_s, \theta_v, q_i$ | Geodesic (§3.4) | FV | Z-grid |
| DYNAMICO | H/NH | $\mathbf{v}_h, \rho_s w, \rho_s, \rho_s\theta_v, \Phi, \rho_s q_i$ | Geodesic (§3.4) | FV | C-grid |
| FV$^3$ | NH | $\mathbf{u}_h, w, \rho_s, \rho_s\theta_v, \delta\Phi, \rho_s q_i$ | Cubed-sphere (§3.2) | FV | D-grid |
| FVM | NH (D) | $\rho_d, \mathbf{u}_h, w, \theta', q_i$ | Octahedral (§3.6) | FV | A-grid |
| GEM | NH | $\mathbf{u}_h, w, \dot{\zeta}, T_v, p, q_i$ | Yin-Yang (§3.7) | FD | C-grid |
| ICON | NH (D) | $\mathbf{u}_h, w, \rho, \theta_v, \rho q_i$ | Icosahedral triangular (§3.3) | FV | C-grid |
| MPAS | NH | $\rho_d\mathbf{u}_h, \rho_d w, \rho_d, \rho_d\theta_v, \rho_d q_i$ | CCVT (§3.5) | FV | C-grid |
| NICAM | NH | $\rho\mathbf{u}_h, \rho w, \rho, \rho e, \rho q_i$ | Geodesic (§3.4) | FV | A-grid |
| OLAM | NH (D) | $\rho\mathbf{u}_h, \rho w, \rho, \rho\theta_{il}, \rho q_i$ | Geodesic (§3.4) | FV | C-grid |
| TEMPEST | NH | $\mathbf{u}_h, w, \rho, \rho\theta_v, \rho q_i$ | Cubed-sphere (§3.2) | SE | A-grid |

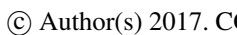



**Table 4.** Vertical staggering (detailed in section 4.1) and vertical coordinates (detailed in section 4.2) for participating dynamical cores.

| Acronym | Vertical Staggering | Vertical Coordinate |
|---|---|---|
| ACME–A | Co-located | floating mass (§4.2.3) |
| CSU | Lorenz | fixed height |
| DYNAMICO | Lorenz | floating mass (§4.2.3) |
| FV$^3$ | Co-located | floating mass (§4.2.3) |
| FVM | Co-located | fixed height |
| GEM | Modified Charney-Phillips (§4.1) | log pressure (§4.2.2) |
| ICON | Lorenz | fixed height |
| MPAS | Lorenz | fixed height |
| NICAM | Lorenz | fixed height |
| OLAM | Lorenz | fixed height with cut-cells (§4.2.4) |
| TEMPEST | Lorenz | fixed height |

**Table 5.** Standard options for diffusion, stabilization, filters, or fixers in participating dynamical cores (detailed in section 6) and temporal discretization (detailed in section 7).

| Acronym | Standard options for diffusion, stabilization, filters, or fixers | Temporal discretization |
|---|---|---|
| ACME–A | 4th-order horizontal hyperviscosity | KGU53 (Guerra and Ullrich, 2016) |
| CSU | 4th-order horizontal hyperviscosity | 3rd-order Adams-Bashforth (AB3) |
| DYNAMICO | 4th-order horizontal hyperviscosity | ARK232 (Giraldo et al., 2013) |
| FV$^3$ | Divergence damping, hyperviscosity | Forward-backwards (Lin and Rood, 1997) / semi-implicit |
| FVM | Monotonic limiting | Semi-implicit (Smolarkiewicz et al., 2014) (§7.2) |
| GEM | Hyperviscosity | Semi-implicit (Girard et al., 2014) (§7.3) |
| ICON | Divergence damping, Smagorinsky, hyperdiffusion | Predictor-corrector |
| MPAS | Smagorinsky, hyperdiffusion | Split-explicit (Klemp et al., 2007) |
| NICAM | 3D divergence damping, Smagorinsky, hyperviscosity | Split-explicit (Klemp et al., 2007) |
| OLAM | Divergence/vorticity damping | 2nd-order Adams-Bashforth, Lax-Wendroff (for tracers) |
| TEMPEST | 4th-order horizontal hyperviscosity | ARS232 (Ascher et al., 1997) |



## 2.2 Colorado State University Model (CSU)

The CSU model uses an optimized geodesic grid to discretize the sphere (Heikes and Randall, 1995; Heikes et al., 2013), with height as the vertical coordinate. The model is based on the non-hydrostatic Unified System of equations proposed by Arakawa and Konor (2009), which filters vertically propagating sound waves but allows the Lamb wave and does not require a reference

5 state. The horizontal wind field is determined by predicting the vertical component of the vorticity and the divergence of the horizontal wind, and then solving a pair of two-dimensional Poisson equations for a stream function and velocity potential. Horizontal diffusion is included in the form of a fourth-order Laplacian operator applied on constant height surfaces ($\nabla_z^4$) that acts on the vorticity, divergence, potential temperature, and tracer. The CSU model supports both third-order and fifth-order upstream-weighted finite-volume advection schemes, with positivity preservation enforced via mass borrowing.

## 10 2.3 DYNAMICO

DYNAMICO is a mimetic finite-difference / finite-volume model, solving initially the hydrostatic primitive equations and recently extended to solve the shallow-atmosphere fully-compressible equations. DYNAMICO's design uniquely combines a representation of the prognostic and diagnostic fields following the ideas of discrete differential geometry, (Dubos et al., 2015). It includes a novel Hamiltonian formulation of the equations of motion in non-Eulerian coordinates (Dubos and Tort,

15 2014) which is imitated at the discrete level using building blocks from the literature (Thuburn et al., 2009; Ringler et al., 2010), and (up to the addition of explicit diffusion) leads to an energy-conserving spatial discretization. It also incorporates a novel explicit-implicit splitting which results in a simple, efficient and scalable implicit solver while allowing stable time steps close or identical to those of the hydrostatic solver (Dubos and Dubey, in preparation). In addition, it features a conservative positive-definite transport scheme based on a slope-limited finite-volume approach (Dubey et al., 2015).

## 20 2.4 FV Cubed (FV$^3$)

The GFDL Finite-Volume Cubed-Sphere Dynamical Core (FV$^3$, or sometimes written FV3) is a fully finite-volume discretization of the fully-compressible non-hydrostatic Euler equations on the equiangular gnomonic cubed-sphere grid with flow-following Lagrangian vertical coordinate (Lin, 2004). The Lagrangian vertical coordinate deforms so that the flow is constrained to follow the Lagrangian surfaces, allowing vertical transport to be represented implicitly without additional ad-

25 vection terms (see section 4.2.3 below). Fluxes are computed using the Piecewise-Parabolic Method of Colella and Woodward (1984) with an optional monotonicity constraint; in non-hydrostatic applications the monotonicity constraint is used primarily for tracer transport. The discretization is on the C-D grid as described by Lin and Rood (1997) (also see section 3.8). Since divergence is effectively invisible to the solver, a divergence damping is applied to control numerical noise as divergent modes cascade to the grid scale. Implicit viscosity is applied through the monotonicity constraint; if non-monotonic advection is used

30 for the momentum and total air mass a weak explicit hyperviscosity is applied for stability and to alleviate numerical noise. Explicit viscosity is applied every acoustic timestep.



The prognostic horizontal winds are stored in the native Gnomonic local coordinate. The non-hydrostatic solver adds a prognostic vertical velocity and geometric height of each grid cell, which can then be used to compute density. All variables are 3D cell-mean values, except for the horizontal winds, which are 2D face-mean values on their respective staggerings; as a result, diagnostic vorticity is a 3D cell-mean value.

## 2.5 Finite-Volume Module (FVM) of the Integrated Forecasting System

The Finite-Volume Module (FVM) of the Integrated Forecasting System (IFS) is currently under development at ECMWF (Smolarkiewicz et al., 2016; Kühnlein and Smolarkiewicz, 2017; Smolarkiewicz et al., 2017). FVM solves the compressible Euler equations in a geospherical framework (Szmelter and Smolarkiewicz, 2010; Smolarkiewicz et al., 2016). A centered two-time-level semi-implicit integration scheme is employed with 3D implicit treatment of acoustic, buoyant, and rotational modes (Smolarkiewicz et al., 2014). The associated 3D Helmholtz problem is solved iteratively using a bespoke preconditioned Generalised Conjugate Residual approach. The integration procedure uses the non-oscillatory MPDATA (Multidimensional Positive Definite Advection Transport Algorithm) advection scheme (Smolarkiewicz and Szmelter, 2005; Kühnlein and Smolarkiewicz, 2017). The horizontal spatial discretization is fully unstructured finite-volume using the median-dual approach. This is combined with a structured-grid finite-difference discretization in the vertical direction. In both the horizontal and the vertical discretization all variables are co-located. The median-dual finite-volume mesh in the horizontal is developed about the nodes of the octahedral reduced Gaussian grid (Section 3.6). The octahedral reduced Gaussian grid is also employed in the spectral-transform dynamical core of the presently operational IFS at ECMWF, which facilitates interoperability of the two formulations. However, FVM is not restricted to this grid and offers capabilities towards broad classes of meshes including adaptivity (e.g. Szmelter and Smolarkiewicz, 2010; Kühnlein et al., 2012).

## 2.6 Global Environmental Multiscale (GEM) model

The GEM model (Girard et al., 2014) is used for operational forecasting at Environment and Climate Change Canada. The horizontal discretization uses the Yin-Yang grid (Kageyama and Sato, 2004) with Arakawa C-grid staggering. The vertical coordinate is a unique hybrid terrain-following coordinate of a log-hydrostatic-pressure type and the vertical discretization is based on the Charney-Phillips grid (see section 4.1). A two time level semi-Lagrangian implicit time discretization is implemented as described in section 7.3. It gives rise to an *iterative* process where each step requires the solution of a linear system of equations that is reduced to a Helmholtz problem for one composite variable. For this problem, a direct solver is involved, using the Schwarz-type domain decomposition method (Qaddouri et al., 2008). The dynamics and physics are time split. To eliminate numerical noise, an explicit hyperviscosity is employed for wind components and tracers via applications of the Laplacian operator, applied after the completion of the physics time step.



## 2.7 Icosahedral Non-hydrostatic model (ICON)

The ICON model (Zängl et al., 2015) discretizes the compressible equations for a shallow atmosphere in vector invariant form for the horizontal wind on a triangular Arakawa C-grid and a smoothed terrain-following height-based Lorenz grid. Prognostic horizontal velocities are stored as normal wind components $v_n$ at the edge mid-points of full levels. Prognostic vertical wind $w$ is stored at the circumcenters of the triangles on half levels. The discretization employs a two time level predictor corrector scheme, which is explicit in all terms except for those describing the vertical propagation of sound waves. Time splitting is applied between the dynamics that is forced by slow physics on the one hand and horizontal diffusion, tracer transport, and fast physics. One complete time step typically includes 5 dynamical sub-steps. The average air mass flux of the dynamical sub-steps is provided to the tracer transport to allow for a mass-consistent transport. For stabilization of the divergence term on the triangular C-grid the divergence in a triangle is computed from modified normal wind components resulting from a weighted average including normal winds on edges of adjacent cells. Further divergence damping is applied to the normal wind at every sub-step. Rayleigh damping is applied to the vertical wind in layers close to the model top in order to avoid the reflection of gravity waves. The horizontal diffusion, which is applied at full model time steps, combines a flow dependent Smagorinsky scheme with a background 4th-order Laplacian diffusion operator. For tracer transport a flux form semi-Lagrangian scheme with monotone flux limiters is used, which grants local mass conservation and consistency with the air motion. The numerical methods have been chosen for high numerical efficiency, and they rely on next-neighbor communication only, thus allowing massive parallelization.

## 2.8 Model for Prediction Across Scales (MPAS)

The Model for Prediction Across Scales (MPAS) (Skamarock et al., 2012) uses an Arakawa C-grid built on a centroidal Voronoi tessellation (see section 3.5), in conjunction with the mimetic TRiSK discretization (Thuburn et al., 2009; Ringler et al., 2010). Advection terms are nominally third- to fourth-order and are handled in accordance with Skamarock and Gassmann (2011). In the vertical, MPAS employs a Lorenz-type second-order nodal finite volume method with a smoothed terrain-following height coordinate. The prognostic variables are dry air pseudodensity $\tilde{\rho}_d$, dry momentum $\tilde{\rho}_d \mathbf{u}$, and a modified moist potential temperature. Integration in time is handled via the split-explicit method of Klemp et al. (2007). Various filters are available for controlling spurious oscillations, including Smagorinsky-type eddy viscosity, and fourth-order hyperdiffusion.

## 2.9 Non-hydrostatic ICosahedral Atmospheric Model (NICAM)

NICAM uses the finite-volume discretization of the fully compressible non-hydrostatic Euler equations on a geodesic grid optimized by the spring dynamics method proposed by Tomita et al. (2002). The system of equations is transformed to the standard terrain-following height coordinate system (Tomita and Satoh, 2004). Instead of the temperature or the potential temperature, the total energy is chosen as a prognostic variable, following the method of (Satoh, 2002, 2003). All of the prognostic variables are collocated horizontally at the mass centroid of each hexagonal/pentagonal cell to mitigate accuracy reduction under cell averaging, which is required in converting cell integrated quantities to point values at cell centroids. The use





of cell centroids ensures quasi second-order accuracy of the gradient and divergence operators of NICAM (Tomita et al., 2001). The Arakawa C-grid staggering is used vertically. A two-stage Runge-Kutta scheme is usually used to reduce computational burden although a three-stage Runge-Kutta scheme (Wicker and Skamarock, 2002) is recommended. The split-explicit time discretization is used for the horizontally propagating sound waves with the divergence damping term (Skamarock and Klemp,

1992). The implicit time discretization is adopted for the vertically propagating one. A variant of the piecewise linear transport scheme (Miura, 2007; Niwa et al., 2011) is used with a flux limiter of Thuburn (1997) for passive tracer transports.

## 2.10 Ocean-Land-Atmosphere Model (OLAM)

OLAM (Walko and Avissar, 2008a, b, 2011) is a deep-atmosphere, fully-compressible model that solves the equations of motion in finite-volume momentum conservation form. Acoustic modes are solved explicitly in the horizontal and implicitly

in the vertical by means of time splitting. Equations are discretized with a C-grid formulation on a hexagonal Voronoi mesh with optional local mesh refinement (which introduces some pentagons and heptagons). Height is the vertical coordinate, and a Lorenz vertical grid staggering is used. A unique feature of OLAM is that grid levels are horizontal and intersect topography. This avoids a number of well-documented errors associated with terrain-following grids and also eliminates the need for evaluation of coordinate transformation terms. Topography is represented as a smooth (non-stepped) surface by means of cut

cells whose surfaces and volume are reduced according to the portion of each cell that is below ground. The OLAM cut-cell formulation conserves mass and momentum.

## 2.11 TEMPEST

The Tempest model (Ullrich, 2014a; Guerra and Ullrich, 2016) is an experimental testbed for high-performance numerical methods that uses a horizontal spectral element discretization and vertical nodal finite volume method with Lorenz staggering

based on the cubed-sphere grid with terrain-following height-based coordinate. The implementation includes both fully explicit time integration, and a horizontally-explicit vertically-implicit formulation that is solved with a third-order implicit-explicit additive Runge-Kutta scheme from Ascher et al. (1997).

## 3 Horizontal discretization and model grids

The horizontal discretization determines how the atmosphere, which consists of a set of approximately continuous fields, is

25 mapped into a very limited and discrete computational space. The horizontal discretization essentially consists of two major choices: the model grid, which determines the density and connectivity of discrete regions (Staniforth and Thuburn, 2012), and the arrangement of prognostic and diagnostic variables around each grid region (Arakawa and Lamb, 1977). In order to meet demands for high computational efficiency and equal partitioning of computation across large parallel systems, modern dynamical cores have explored a number of options for model grids. The choice of model grid can be motivated by simplicity,

as in the case of the latitude-longitude grid, by a desire to maintain a local Cartesian structure, as with the cubed-sphere grid, or to support grid isotropy and homogeneity, as with many of the hexagonal or Voronoi grids that have been employed. The choice




of grid may be further decided by the numerical method – for instance, finite element models that use tensor products to define basis functions require grids consisting entirely of quadrilaterals. Inevitably a choice must be made, and the pros and cons of that choice will impact other decisions related to the model. To better understand the options that are available to dynamical core developers, we begin by reviewing many of the model grids that have been employed in global dynamical cores around the world. Then, in section 3.8, we discuss the "staggering" of model variables, referring to the distribution of variables within and around each grid cell.

### 3.1 Latitude-longitude grid

The classic latitude-longitude grid consists of a subdivision of the sphere produced by subdividing along lines of constant latitude and longitude. Because of the convergence of grid lines near the poles, the operational use of this grid requires that the associated numerical scheme be resilient to arbitrarily small Courant number, or that polar filtering be employed to remove unstable computational modes (Lin, 2004). This grid is employed by the UK Met Office (Davies et al., 2005; Wood et al., 2014).

### 3.2 Cubed-sphere grid

The equiangular gnomonic cubed-sphere grid (Sadourny, 1972; Ronchi et al., 1996; Putman and Lin, 2007) consists of six Cartesian patches arranged along the faces of a cube which is then inflated onto a spherical shell. More information on this choice of grid can be found in Ullrich (2014a). On the equiangular cubed-sphere grid, coordinates are given as $(\alpha, \beta, p)$, with central angles $\alpha, \beta \in [-\frac{\pi}{4}, \frac{\pi}{4}]$ and panel index $p$. The structure of this grid supports refinement through stretching (Schmidt, 1977; Harris et al., 2016) or nesting (Harris and Lin, 2013). The Cartesian structure of cubed-sphere grid panels is advantageous for numerical methods that are formulated in Cartesian coordinates, or that utilize dimension splitting. Nonetheless, special treatment of the panel boundaries is often necessary since they represent coordinate discontinuities. This grid is depicted in Figure 1 (left).

### 3.3 Icosahedral (triangular) grid

The icosahedral triangular grid is derived from the spherical icosahedron that consists of 20 equilateral spherical triangles, 30 great circle edges and 12 vertices. These initial triangles are then subdivided repeatedly until the desired mean resolution is obtained. For a single subdivision each edge is divided in $n$ arcs of equal length, thus defining new vertices, which by proper connection to other new vertices result in $n^2$ triangles filling the original triangle. By construction the new vertices share 6 triangles, thus the refinement process brakes the initial isotropy of the icosahedron and results in non-equilateral triangles of different sizes.

Several methods are available for subdividing the triangular regions. One such approach is implemented by the ICON grid generator, which allows an "arbitrary" subdivision factor $n$ for the first refinement step only, the so-called root refinement. Typical choices are $n = 2, 3$ or $5$. All additional $m$ refinement steps use $n = 2$, i.e. are bisection steps. A global grid resulting




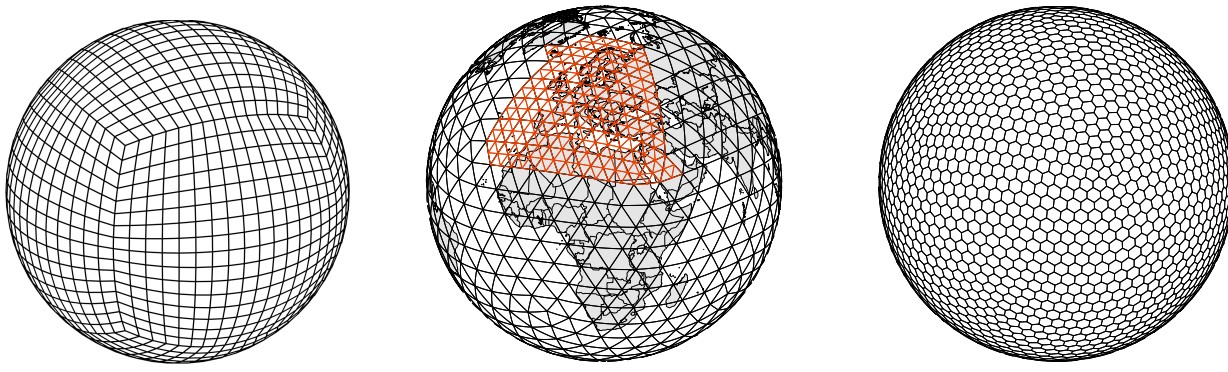

**Figure 1.** (Left) A cubed-sphere grid. (Center) An icosahedral (triangular) grid with additional refinement over Europe, as indicated in red. (Right) An icosahedral (hexagonal) grid.

from a root division factor $n$ and $m$ bisections, denominated as *RnBm* grid, has $n_c = 20 \cdot n^2 \cdot 2^{2m}$ cells, $n_e = 3/2 \cdot n_c$ edges and $n_v = 10 \cdot n^2 \cdot 2^{2m} + 2$ vertices. The anisotropy of global grids is reduced by the spring dynamics of Tomita et al. (2001). An example of such a grid is depicted in Figure 1 (center). A discussion of the effective resolution of such grids is given in Dipankar et al. (2015). The ICON grid generator further allows for inset regional grids, produced by additional refinement steps

that are only applied over a limited region, or set of regions. The dynamical core then allows for either one-way or two-way coupling of the refined region to the parent model. The current operational numerical weather prediction of the Deutscher Wetterdienst (German Weather Service, DWD) for instance uses a $R3B7$ global grid with 2949120 cells and 13 km mean resolution in combination with a refined region over Europe at 6.5 km resolution.

### 3.4   Icosahedral (hexagonal) grid / geodesic grid

The icosahedral (hexagonal) grid, also commonly referred to as the geodesic grid, is most directly obtained by taking the dual to the icosahedral (triangular grid) – that is, by replacing grid nodes with spherical polygons. The resulting grid's cells are hexagonal, except for twelve pentagonal cells. Given an icosahedral-triangular mesh, vertices of the corresponding icosahedral-hexagonal mesh are then defined as either circumcenters or barycenters of triangles, leading to either a Voronoi mesh, used by DYNAMICO (see also section 3.5), or a barycentric mesh, used by NICAM. A Voronoi mesh has the property that triangular

edges are perpendicular to edges of hexagons/pentagons, facilitating the formulation of certain finite-difference and finite-volume numerical schemes. The resulting highly homogeneous and isotropic grid then appears analogous to the grid in Figure 1 (right). Unlike the cubed-sphere and icosahedral (triangular) grid, grid cells on this geodesic grid are guaranteed to be edge-neighbors (cells that share a given edge) if they are also node-neighbors (cells that share a given node).

It is often useful to optimize icosahedral-hexagonal grids as well. DYNAMICO applies a number of iterations of Lloyds algo-

rithm (Lloyd, 1982), following by replacing the vertices of the original triangular mesh by the centroid of hexagons/pentagons,

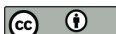



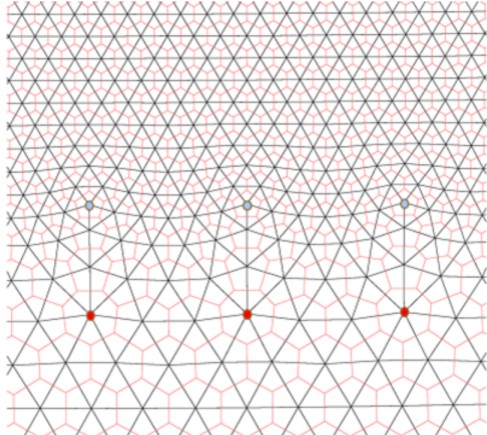

**Figure 2.** Detail of one step of local mesh refinement used by OLAM Voronoi mesh. The transition zone is constructed by explicit topological reconnection of the grid lines, which produces pairings of heptagons (red dots) and pentagons (blue dots) along the refinement perimeter.

then re- generating the icosahedral-hexagonal mesh. This improves the homogeneity of the grid (e.g. ratio of largest cell area to smallest cell area) but several thousand iterations can be required for a significant improvement.

OLAM optimizes by applying the spring dynamics method of Tomita et al. (2001) to the dual triangular mesh prior to its mapping to the Voronoi grid. When local mesh refinement is applied, which OLAM achieves in a series of one or more resolution-doubling steps, each spanning a transition zone that is three grid rows wide (Figure 2), the equilibrium spring length is scaled to the target grid cell size in each refinement level and is varied incrementally across the transition zone. Spring dynamics is further modified by forcing angles on the dual triangular mesh in the transition zone in order to move the triangle edges closer to the centers of the hexagon edges they intersect.

### 3.5 Constrained Centroidal Voronoi Tessellation (CCVT) grids

Given a set of $N$ distinct points on the sphere $x_i$ (referred to as the generators, $1 \leq i \leq N$), the *Voronoi tessellation* (or the *Voronoi diagram*) associated with the generators is the set of polygons $\Omega_i$ consisting of all points that are closer (in the sense of great-circle distance) to $x_i$ than any other $x_j$ with $i \neq j$ (Okabe et al., 2009). For a given set of generators, this tiling is unique and completely covers the sphere, and so can be employed in conjunction with many finite volume methods. However, for an arbitrary set of generators it is easy to produce highly distorted polygons, particularly if the density of generators varies substantially. This has led to the development of *constrained centroidal Voronoi tessellation (CCVT)* (Du et al., 2003), which imposes the additional requirement that the set of generators be coincident with the centroids of each polygon. Given a desired polygonal density function, several algorithms have been developed to generate CCVTs both in Cartesian and spherical geometry (i.e. for ocean basins or ice sheets) (Ringler et al., 2008). Figure 3 depicts one such CCVT grid that is compatible with the MPAS model. CCVT grids are often confused with deformations of the icosahedral (hexagonal) grid described in section





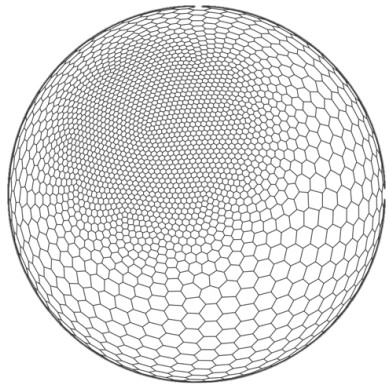

**Figure 3.** A constrained centroidal Voronoi tessellation grid with localized grid density that could be employed in the MPAS model.

3.4, since both typically contain a large number of hexagonal elements. However, CCVT grids are fundamentally constructed using a very different technique and will often contain polygonal elements with more than six sides.

### 3.6 Octahedral reduced Gaussian grid

As with the classical reduced Gaussian grid of Hortal and Simmons (1991), the octahedral reduced Gaussian grid (Malardel
et al., 2016; Smolarkiewicz et al., 2016) specifies the latitudes according to the roots of the Legendre polynomials. The two grids differ in the arrangement of the points along the latitudes, which follows a simple rule for the octahedral grid: starting with 20 points on the first latitude around the poles, four points are added with every latitude towards the equator, whereby the spacing between points along the latitudes is uniform and there are no points at the equator. The octahedral reduced Gaussian grid is suitable for transformations involving spherical harmonics, and has been introduced for operational weather prediction
with the spectral dynamical core of the IFS at ECMWF in 2016. Figure 4 depicts the octahedral reduced Gaussian grid nodes together with the edges of the primary mesh as applied in the context of the finite-volume discretization of FVM (Section 2.5).

### 3.7 Yin-Yang grid

The overset Yin-Yang grid (Kageyama and Sato, 2004) has two Cartesian grid components (subsets of a latitude-longitude grid) which are geometrically identical (see Figure 5). These components are combined to cover a spherical surface with
15 partial overlap along their borders. The Yin component covers the latitude-longitude region

$$(-\frac{\pi}{4} - \delta_\theta \leq \theta \leq \frac{\pi}{4} + \delta_\theta) \cap (-\frac{3\pi}{4} - \delta_\lambda \leq \lambda \leq \frac{3\pi}{4} + \delta_\lambda), \tag{1}$$

where $\delta_\lambda, \delta_\theta$ are small buffers that are proportional to the respective grid-spacings and are required to enforce a minimum overlap in the overset methodology. For instance, a common configuration employed by the GEM model for DCMIP fixes $\delta_\theta = 2$ degrees and $\delta_\lambda = 3\delta_\theta$. The Yang component covers an analogous area, but is rotated perpendicularly so as to cover the





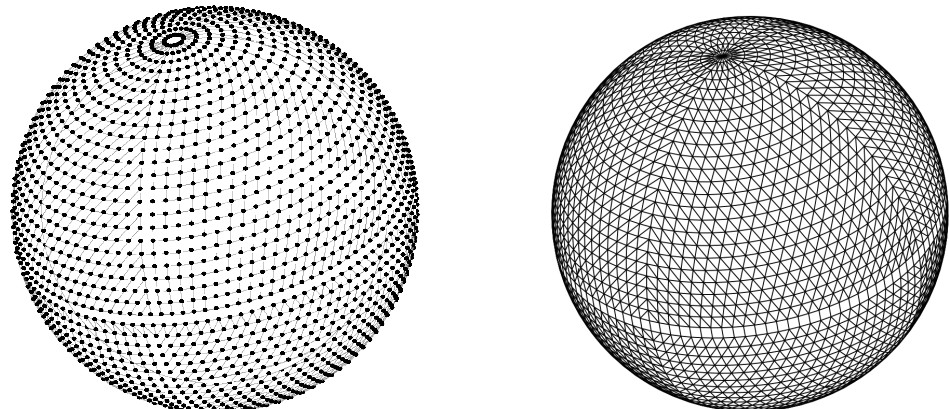

**Figure 4.** Locations of the octahedral reduced Gaussian grid nodes (left), and the edges of the primary mesh connecting the nodes as applied with the finite-volume discretisation in FVM (right). A coarse octahedral grid with only 24 latitudes between pole and equator 'O24' is used for illustration. The dual mesh resolution of the octahedral reduced Gaussian grid is about a factor 2 finer at the poles than the equator, see Smolarkiewicz et al. (2016).

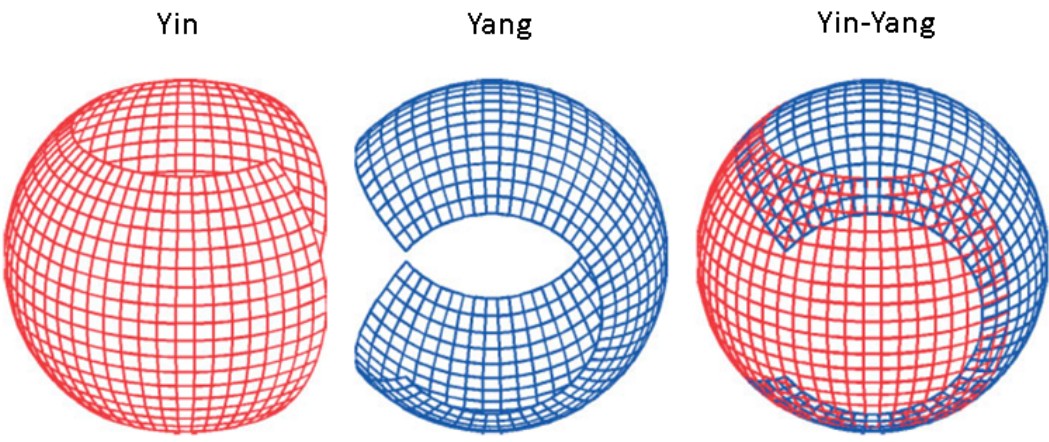

**Figure 5.** The Yin-Yang grid is a combination of two limited-domain latitude-longitude grids assembled to provide complete coverage of the sphere.

region of the sphere outside of the Yin grid. This grid is employed by the GEM model, utilizing a pair of regional climate models on the two Cartesian patches.

## 3.8 Horizontal staggering

The horizontal placement of variables impacts a number of properties of the numerical method, including how energy and

5  enstrophy conservation is managed, any computational modes that might arise due to differencing, dispersion properties, and



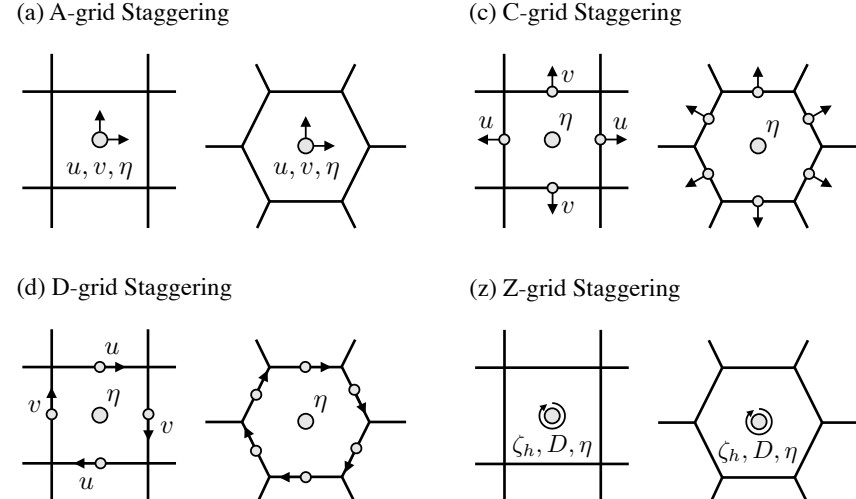

**Figure 6.** Horizontal staggering options represented among DCMIP models, in this case depicted on a rectilinear grid and geodesic grid. Here $\eta$ denotes the buoyancy variable.

the maximum stable timestep size (Randall, 1994; Ullrich, 2014b). The original four Arakawa grids (Arakawa and Lamb, 1977), denoted with letters A- through D- were initially designed for rectilinear meshes, but were later adapted for a variety of unstructured grids. Later, other grid types were added, including the Z-grid, which used the vertical component of vorticity and the horizontal divergence in place of the velocity components (Randall, 1994), and the ZM-grid, which extends the B-
grid to hexagons by placing the velocity at hexagonal nodes (Ringler and Randall, 2002). By interpreting "staggerings" to be analogous to a choice of finite element basis, new staggerings are under development in the context of mixed finite element methods (Cotter and Shipton, 2012). Among the models that participated in DCMIP, only four grids were represented: The A-grid, which involves simple co-location of all velocity components; the C-grid, which places perpendicular velocity components on grid edges; the D-grid, which places parallel velocity components on grid edges; and the Z-grid (see Figure 6).
Arguments in favor or against particular staggerings have generally emerged from linear analyses, and typically in the absence of either implicit or explicit diffusion. In this context, the A-grid tends to support large time step sizes, but produces unphysical phase speeds and negative group velocities at high wavenumbers, including a stationary $2\Delta x$ wavelength mode (even in the context of finite element methods); the C-grid better represents short wave modes, does not support extraneous computational modes (as long as the number of edges is equal to twice the number of volumes), but also requires a timestep that
is half that of the A-grid (in the 2nd order case), or 40% smaller (in the 4th order case) (Ullrich, 2014b); the D-grid provides a better representation of vorticity, but produces unphysical effects analogous to those on the A-grid at high wavenumbers that must be controlled with divergence damping; finally, the Z-grid yields optimal dispersion properties, but requires the inversion of a Helmholtz problem at each timestep to extract the velocity field from the divergence and vorticity.



Other specialized staggerings have been developed that couple horizontal staggering with the formulation of the time integrator. In the FV$^3$ model, although velocities are stored in accordance with the D-grid arrangement, at the intermediate stages of the forward-backwards timestepping scheme, velocities are actually prognosed on the C-grid. The intermediate velocities then act as a simplified Riemann solver: the intermediate stage velocities are time-centered and can be used to compute the

fluxes and advance the flux terms by a full acoustic timestep. More details on this approach can be found in Lin and Rood (1997).

## 4  Vertical discretization

Because of the vast differences between horizontal and vertical scales in global simulations, most atmospheric models use dimension splitting in order to separate the horizontal discretization from the vertical discretization. In this section, design

considerations related to the vertical column are discussed, including the staggering of prognostic and diagnostic variables, and the choice of vertical coordinate.

### 4.1  Vertical staggering

Along with the choice of prognostic variables, the vertical discretization of the equations of motion also allows for the staggered placement of prognostic variables. As with hydrostatic models, certain discretizations give rise to spurious computational modes that can contaminate the solution (Tokioka, 1978; Arakawa and Moorthi, 1988). The choice of vertical staggering may

also impact many physically-relevant properties of the model near the grid scale, such as the phase speed of Rossby waves (Thuburn and Woollings, 2005). Finally, the choice of vertical staggering can have impacts on the physics-dynamics coupling Holdaway et al. (2013a, b). Taken altogether, these issues suggest care should be taken when selecting the discretization. Since co-located discretizations of the non-hydrostatic equations generally require some additional effort to control spurious com-

putational modes, it is more common to employ either: (a) a Lorenz-type staggering (Lorenz, 1960), which places horizontal velocity, buoyancy, and thermodynamic variables on model levels, and vertical velocity on model interfaces; or (b) a Charney-Phillips-type staggering (Charney and Phillips, 1953), which places horizontal velocity and buoyancy variables on model levels and vertical velocity and thermodynamic variables on model interfaces (see Figure 7). These approaches can be further augmented as needed, for instance by shifting the vertical velocity and thermodynamic variables from the bottom boundary to an

intermediate level, as in the GEM model. Note that, in general, tracer variables are co-located with the buoyancy variable.

### 4.2  Vertical coordinates

In the context of dimension splitting, the "horizontal" typically refers to either the contravariant basis, which is perpendicular to the vertical, or the covariant basis, which is directed along coordinate (e.g. terrain-following) surfaces. In contrast, the vertical dimension is strictly aligned with the radial vector pointing from the center of the Earth. Vertical position is typically labelled

using an arbitrary function $s(t, \mathbf{x}, z)$ that is monotonic in $z$, so that model interfaces are equally spaced with respect to $s$. Typically $s$ is chosen so that the Earth's surface (the bottom boundary of the atmosphere) is a coordinate surface, allowing for





**Figure 7.** (a) A Lorenz-type variable staggering for a model utilizing height coordinates, (b) a Charney-Phillips-type variable staggering for a model utilizing height coordinates, (c) a modified Charney-Phillips-type staggering used in the GEM model that introduces a new near-surface level for vertical velocity and temperature.

easy specification of boundary conditions for the prognostic equations – this leads to the so-called "terrain-following" family of vertical coordinates. Perhaps the most common terrain-following coordinate is from Gal-Chen and Somerville (1975), which is in terms of the altitude $z$ and takes the form

$$s(\mathbf{x}, z) = z_{\text{top}} \left[ \frac{z - z_s(\mathbf{x})}{z_{\text{top}} - z_s(\mathbf{x})} \right], \tag{2}$$

where $\mathbf{x}$ denotes the horizontal position, $z_s(\mathbf{x})$ is the height of the topography at that position, and $z_{\text{top}}$ denotes the height of the model top (typically independent of position). Analogous formulations are available for mass-based ($\sigma$-coordinates) and entropy-based vertical coordinates. Because the sharp variations in the coordinate surfaces are preserved far above a rough lower-boundary, new coordinate formulations have been proposed that smooth coordinate surfaces, such as Schär et al. (2002) or Klemp (2011). All models in this paper except for OLAM use some variant of terrain-following coordinates, although work on developing modern cut-cell, embedded boundary and immersed boundary representations is ongoing (e.g. Lock et al. (2012)). Note that time-dependent vertical coordinates are allowed and are typically referred to as "floating" coordinates. Several examples of vertical coordinates are now given.

### 4.2.1 Mass-based coordinates

Mass-based coordinates (Laprise, 1992) are a generalization of pressure-based coordinates to non-hydrostatic models, with a vertical coordinate defined as the total gravity-weighted overhead mass,

$$s = \int_z^\infty \rho g \, dz. \tag{3}$$





Under this definition,

$$\frac{\partial s}{\partial z} = -\rho g. \tag{4}$$

### 4.2.2 GEM zeta coordinate

The vertical coordinate in the GEM model, denoted $\zeta$, is a hybrid terrain-following coordinate of a log-hydrostatic-pressure
type. Taking $s$ (denoted $\pi$ in GEM documentation) as given in (3), then $\zeta$ is given by the relation

$$\log s = A(\zeta) + B(\zeta)\left[\log s(z_s) - \zeta_s\right], \tag{5}$$

with

$$A(\zeta) = \zeta, \quad \text{and} \quad B(\zeta) = \left(\frac{\zeta - \zeta_{top}}{\zeta_s - \zeta_{top}}\right)^r. \tag{6}$$

Here $\zeta_s = \log(10^5)$, $\zeta_{top} = \log(s_{top})$, $s_{top}$ is the coordinate value at the uppermost interface, and $r$ is a variable exponent
providing added freedom for adjusting the thickness of model layers over high terrain.

### 4.2.3 Floating Lagrangian coordinates (ACME–A, DYNAMICO and FV$^3$)

In the floating Lagrangian formulation (Starr, 1945; Lin, 2004) the vertical coordinate is chosen to represent an artificial tracer
with monotonically increasing or decreasing mixing ratio $s$ in the vertical. The actual mixing ratio at initiation is arbitrary, and
can be constructed to be height-like (i.e., $s = z$), or mass-like, i.e.

$$s = \int_z^\infty \rho_0 g \, dz, \tag{7}$$

in which case a 3D reference density field $\rho_0$ can be imposed. Of primary importance is the fact that the vertical coordinate
satisfy

$$\dot{s} = \frac{ds}{dt} = 0, \tag{8}$$

which greatly simplifies the associated prognostic velocity and continuity equations. Floating Lagrangian coordinates are often
paired with a vertical remapping operation that corrects for strong grid distortions that may occur after sufficiently long model
integrations.

### 4.2.4 Cut-cells in OLAM

A pure z coordinate with horizontal grid levels is used in OLAM (Walko and Avissar, 2008b) in order to completely avoid
topographic imprinting on the model grid levels (Figure 8). This implies that grid levels intersect the topographic surface,
leading to some grid cells being partially above and partly below the surface. The face areas of these so-called cut cells are
reduced accordingly, which in turn regulates cell-to-cell flux transport in accordance with the kinematic constraint imposed
by the topography. Cut cell volumes are also reduced, and volumes and surface areas of all cells appear explicitly in the
finite-volume formulation of the mass and momentum conservation equations.





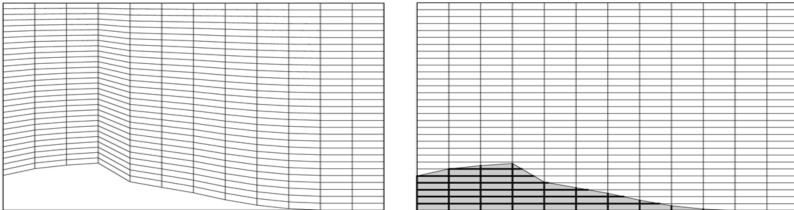

**Figure 8.** (Left) A terrain-following coordinate passing over rough topography. (Right) A cut-cell coordinate used for representing the same topography.

## 5   Prognostic equations and treatment of moisture

The Naviér-Stokes equations that govern atmospheric motion can take on many forms, depending on the choice of prognostic variables and coordinate system. A derivation of many forms of these equations can be found in Appendix A. The particular prognostic equations used by the model can impact the presence of computational modes, the accuracy of the model in representing the physical modes of the atmosphere (Thuburn and Woollings, 2005), and the ability of the model to conserve important invariants such as energy (Dubos and Tort, 2014). The remainder of this section gives some specific examples of prognostic equations used by the DCMIP models, including any special treatment of terms related to moist physics.

### 5.1   ACME–A

ACME-A hosts an experimental compressible shallow-atmosphere model in hybrid terrain-following pressure vertical coordinates $\eta$, similar to the model of Laprise (1992). The 2D vector invariant form of the prognostic horizontal velocity equations (A62) is employed, in conjunction with prognostic potential temperature (A57), pseudodensity (A55), and geopotential (A27). The vertical velocity equation is formulated analogous to that of GEM,

$$\frac{dw}{dt} = -g_c \left( 1 - \frac{\partial p}{\partial s} \right). \tag{9}$$

### 5.2   DYNAMICO

The prognostic equations employed by DYNAMICO are based on a Hamiltonian formulation (Dubos and Tort, 2014). The specific prognostic variables employed pseudo-density $\rho_s$, mass-weighted tracers (potential temperature, water species), geopotential $\Phi$, horizontal covariant components of momentum and mass-weighted vertical momentum $W = \rho_s g^{-2} d\Phi/dt = \rho_s g^{-1} w$. Prognostic equations are in flux-form for mass (A55) and $W$ (A23), in advective form for $\Phi$ (A27), and in vector-invariant form for covariant horizontal momentum (A76).

### 5.3   FV$^3$

The hydrostatic FV$^3$ model uses a mass-based floating Lagrangian coordinate along with the shallow-atmosphere approximation (Lin, 2004). Prognostic equations include horizontal velocity in 2D vector-invariant form (A38), pseudo-density (A55),





and virtual potential temperature (A57). The non-hydrostatic model further incorporates prognostic geopotential (A27) and vertical momentum (A37).

## 5.4 FVM

The FVM formulation is based on conservation laws for dry mass (10a), momentum (10b), and dry entropy (10c) in Eulerian flux-form, which are similar to (A9) for $\rho_d$, (A23), and (A13) for $\theta$, respectively. Moreover, underlying the conservation laws in FVM is a perturbational form with respect to a balanced ambient state and a generalized curvilinear coordinate formulation in a geospherical framework. Following Smolarkiewicz et al. (2017), the FVM governing equations can concisely be written as

$$\frac{\partial \mathcal{G}\rho_d}{\partial t} + \overline{\nabla} \cdot (\mathbf{v}\mathcal{G}\rho_d) = 0 \,, \tag{10a}$$

$$\frac{\partial \mathcal{G}\rho_d\mathbf{u}}{\partial t} + \overline{\nabla} \cdot (\mathbf{v}\mathcal{G}\rho_d\mathbf{u}) = \mathcal{G}\rho_d \left( -\theta_\rho \widetilde{\mathbf{G}}\overline{\nabla}\phi' - \mathbf{k}\,g\left( \frac{\theta'}{\theta_a} + \varepsilon_b\,q_v' - q_c - q_r \right) - 2\boldsymbol{\Omega} \times \left( \mathbf{u} - \frac{\theta_\rho}{\theta_{\rho a}}\mathbf{u}_a \right) + \mathcal{M}' \right) \,, \tag{10b}$$

$$\frac{\partial \mathcal{G}\rho_d\theta'}{\partial t} + \overline{\nabla} \cdot (\mathbf{v}\mathcal{G}\rho_d\theta') = -\mathcal{G}\rho_d\widetilde{\mathbf{G}}^T\mathbf{u} \cdot \overline{\nabla}\theta_a \,, \tag{10c}$$

$$\phi' = c_{pd} \left[ \left( \frac{R_d}{p_0}\rho_d\theta\left(1 + q_v/\varepsilon\right) \right)^{R_d/c_{vd}} - \pi_a \right] \,. \tag{10d}$$

Dependent variables in (10) are dry density $\rho_d$, 3D physical velocity vector $\mathbf{u}$, potential temperature perturbation $\theta'$, and a modified Exner pressure perturbation $\phi'$, with the thermodynamic variables related by the gas law (10d). Primes indicate perturbations with respect to the prescribed ambient state denoted by subscript 'a', see Prusa et al. (2008) and Smolarkiewicz et al. (2014) for discussions. The symbol $g$ in (10b) denotes the gravitational acceleration and $\varepsilon_b = 1/\varepsilon - 1$. As far as geometric aspects are concerned, the nabla operator $\overline{\nabla}$ represents the 3D vector of partial derivatives with respect to the curvilinear coordinates, along with the Jacobian $\mathcal{G}$, a matrix of metric coefficients $\widetilde{\mathbf{G}}$, its transpose $\widetilde{\mathbf{G}}^T$, and the contravariant velocity $\mathbf{v} = \widetilde{\mathbf{G}}^T\mathbf{u}$ where a contribution from optional time-dependency of the curvilinear coordinates is neglected for simplicity (Kühnlein and Smolarkiewicz, 2017). The symbol $\mathcal{M}' = \mathcal{M}'(\mathbf{u}, \mathbf{u}_a, \theta_\rho/\theta_{\rho a})$ in (10b) subsumes the metric forces in the spherical domain (Smolarkiewicz et al., 2017).

## 5.5 GEM

In GEM the non-hydrostatic equations are written explicitly as deviations from hydrostatic balance represented by

$$\mu = \frac{\partial p}{\partial s} - 1, \tag{11}$$



where $s$ (denoted $\pi$ in GEM documentation) is given by (3). In this case the equations of GEM model (Girard et al., 2014) are concisely given by

$$\frac{d\mathbf{u}_h}{dt} + f\mathbf{k} \times \mathbf{u_h} + R_d T_v \nabla_\zeta \log p + (1+\mu)\nabla_\zeta \Phi = 0, \tag{12}$$

$$\frac{dw}{dt} - g_c \mu = 0, \tag{13}$$

$$\frac{d}{dt}\log\left(\frac{\partial s}{\partial \zeta}\right) + \nabla_\zeta \cdot \mathbf{u}_h + \frac{\partial \dot{\zeta}}{\partial \zeta} = 0, \tag{14}$$

$$\frac{d\log T_v}{dt} - \frac{R_d}{c_p}\frac{d\log p}{dt} = 0, \tag{15}$$

$$\frac{\partial \Phi}{\partial s} + \frac{R_d T_v}{p} = 0, \tag{16}$$

$$\frac{d\Phi}{dt} - g_c w = 0. \tag{17}$$

Here $\nabla_\zeta$ denotes the horizontal gradient along $\zeta$ surfaces. With respect to the treatment of moisture in GEM, the cloud water and all non-gases are embedded in the total air density $\rho$, affecting the virtual temperature defined in (A7). Also, specific mass is used in GEM (not mixing ratio).

### 5.6 ICON

ICON solves a non-hydrostatic equation set based on Gassmann and Herzog (2008) using terrain-following z-coordinates. The governing equations describe the mixture of a two-component system of dry air and water, where water is allowed to occur in all three phases, including precipitating constituents. Following Wacker et al. (2006), the barycentric (bc) velocity $\mathbf{u}_{bc} = \sum_k \rho_k \mathbf{u}_k / \sum_k \rho_k$ – that is the mass-weighted sum of all constituent-specific velocites (including sedimenting ones) – is used as a prognostic variable. In contrast to Gassmann and Herzog (2008), a vector invariant form is only used for the horizontal velocity equation (A33), whereas the vertical velocity equation is solved in advective form. The pressure gradient force is formulated according to (A20).

Additional prognostic variables include total air density (A10), virtual potential temperature (A57), and mass fractions $q_k = \rho_k/\rho$ of all constituents (except for dry air) for which the prognostic equation reads

$$\frac{\partial \rho q_k}{\partial t} + \nabla \cdot (\rho q_k \mathbf{u}_{bc}) = -\nabla \cdot \mathbf{J}_k + \sigma_k, \tag{18}$$

with $\sigma_k$ describing sources/sinks due to phase changes, and $\mathbf{J}_k = \rho q_k (\mathbf{u}_k - \mathbf{u}_{bc})$ denoting diffusion fluxes, which account for the motion of constituents relative to the frame of reference set by $\mathbf{u}_{bc}$.

The specific heat capacities and ideal gas constant are approximated to be equal to their dry values $R^* \approx R_d$, $c_p^* \approx c_{pd}$, $c_v^* \approx c_{vd}$. The model also uses a prognostic equation for Exner pressure to simplify the treatment of vertical sound wave propagation, given by

$$\frac{\partial \pi}{\partial t} + \frac{R_d}{c_{vd}}\frac{\pi}{\rho \theta_v}\nabla \cdot (\mathbf{u}_{bc}\rho\theta_v) = \hat{Q}, \tag{19}$$



where $\hat{Q}$ is an appropriately formulated diabatic heat term. The horizontal uses a Arakawa C-grid formulation on the triangular grid to prognose horizontal velocities normal to triangle edges $v_n$, making use of reconstructed tangential velocity components $v_t$.

In the current implementation, the following simplifcations are made with regards to the treatment of moisture: The atmo-spheric mass loss/gain due to precipitation/evaporation is neglected in the total mass continuity equation (A10), by setting the vertical component of $\mathbf{u}_{bc}$ to zero at the lower boundary: $w_{bc}|_{sfc} = 0$. In addition, only the vertical diffusion fluxes $\mathbf{J}_k$ of sedimenting constituents and the surface evaporation flux $\mathbf{J}_v|_{sfc}$ are taken into account. The counter-flux of non-sedimenting constituents is discarded. Since in the given framework the continuity equation (A10) is only valid if the constraint $\sum_k \mathbf{J}_k = 0$ holds, we (implicitly) assume a fictitious counter-flux of dry air to compensate for the considered vertical diffusion fluxes. As a consequence, ICON currently conserves the global integral of total air mass rather than dry air mass.

## 5.7 OLAM

OLAM solves the deep-atmosphere, fully-compressible equations in mass- and momentum-conserving finite-volume form using equations (A10), (A23), and (A13). Prognostic variables are the 3 components of momentum, ice-liquid potential tem-perature $\theta_{il}$ (Walko et al., 2000), total density $\rho$, specific density of total water, and specific bulk density and/or bulk number concentration of various scalar quantities including liquid and ice hydrometeors, aerosols, and trace gases. For DCMIP, the latter are limited to cloud and rain specific bulk densities. Water vapor density is diagnosed by subtracting bulk densities of all liquid and ice hydrometeors from the total water density, dry air density is diagnosed by subtracting total water density from total density, and pressure is diagnosed based on the equation of state and values of dry air density, water vapor density, and potential temperature $\theta$. The latter is in turn diagnosed from $\theta_{il}$ and from the latent heat required to convert any hydrometeors present to the vapor phase. Velocity components are diagnosed by dividing momentum components by total density.

Momentum is C-staggered in the horizontal and vertical (Lorenz vertical staggering is used), meaning that prognosed com-ponents live on the grid cell faces and are each normal to the respective face and the pressure gradient force is evaluated and applied at those locations. However, evaluation of advective and turbulent momentum transport (as well as the Coriolis force) involves a diagnostic reconstruction of the total momentum vector at the centers of scalar grid cells (Perot, 2000), and cell-to-cell flux of momentum is computed from that reconstruction using the same A-grid control volumes as for scalars. This arrangement is particularly convenient for the cut-cell formulation at the topographic surface where reduced cell face areas and volumes regulate momentum and scalar fluxes in an identical manner.

## 5.8 TEMPEST

Tempest is a shallow-atmosphere Eulerian model with terrain-following $z$-coordinates with prognostic density (A55), virtual potential temperature (A57), and vector-invariant form for covariant horizontal velocity (A76) and vertical momentum (A32).





## 6   Diffusion, stabilization, filters and fixers

Most dynamical cores implement specialized techniques for diffusion or stabilization (see Table 5). Diffusion is a numerical technique that removes spurious numerical noise from the simulation, where the numerical noise typically arises because of inaccuracies in the treatment of waves with wavelengths near the grid scale. Stabilization is a numerical technique that prevents

energy growth and allows the model to be run over long periods. Diffusion or stabilization options include physically-motivated turbulence parameterizations, added viscosity or hyperviscosity terms with tunable coefficients, off-centering, or wave-mode filters. Since the discretization can also lead to an unphysical loss of mass or energy, mass or energy fixers are also employed to replace lost mass or energy to the system. A comprehensive overview of schemes for diffusion and stabilization schemes can be found in Jablonowski and Williamson (2011). In this section we discuss some of the diffusion and stabilization strategies

employed by the DCMIP suite of dynamical cores.

### 6.1   ACME–A / TEMPEST

In both ACME–A and Tempest, scalar hyperviscosity is employed for $\rho$, $\theta$ and tracer variables via repeated application of a scalar Laplacian (Dennis et al., 2012; Ullrich, 2014a). Vector hyperviscosity is also applied by decomposing the horizontal vector Laplacian into divergence damping and vorticity damping terms via the vector identity

$$\nabla^2 \mathbf{u}_h = \nabla \nabla \cdot \mathbf{u}_h + \nabla \times \nabla \times \mathbf{u}_h. \tag{20}$$

Both viscosity operations are applied after the completion of all Runge-Kutta sub-cycles. Several limiter options are available for tracer transport including a sign-preserving limiter and a monotone optimization base limiter described in Guba et al. (2014).

### 6.2   FV$^3$

Explicit dissipation in FV$^3$ is applied separately to the divergence and to the horizontal fluxes in the governing equations. The D-grid discretization applies no direct implicit dissipation to the divergence, so divergence damping is an intrinsic part of the solver algorithm since otherwise there are no processes by which energy contained in the divergent modes is removed at the grid scale. FV$^3$ has options for fourth-, sixth-, or eighth-order divergence damping; a second-order option is also available for use in idealized convergence tests, which can be applied in addition to the higher-order diffusion. The monotonicity constraint

used in computing the fluxes in the momentum, thermodynamic, and mass continuity equations is sufficient to damp and stabilize the non-divergent component of the flow. If additional damping is desired, or if the non-monotonic advection is used, there is an option to apply hyperdiffusion to the fluxes in each of these equations, with the exception of the tracer transport, which always uses monotonic transport with no explicit diffusion. The hyperdiffusion is of the same order as but much smaller than the divergence damping. Both divergence damping and hyperdiffusion are applied along the Lagrangian surfaces and are

re-computed every acoustic timestep.





Wave absorption at the model top is also provided by the flexible-lid (constant-pressure) upper boundary; FV³ also applies second-order diffusion to all fields, except the tracers, to create a sponge layer, typically comprising the top two layers of the domain, to damp other signals reaching the top of the domain. An energy-conserving Rayleigh damping, applied consistently to all three components of the winds, is also available, which is strongest in the top layer of the domain and becomes weaker

with distance until reaching a runtime-specified cut-off pressure.

FV³ has an option to restore lost energy by the adiabatic dynamics, in whole or a fraction thereof (decided by a namelist option at runtime), by globally adding a Exner-function weighted potential temperature increment. This is only done before the physics is called and is not used in idealized simulations.

## 6.3    GEM

An explicit hyperviscosity in GEM is handled via applications of the Laplacian operator for both wind components and tracers. A vertical sponge layer, which uses a Laplacian operator, is employed on wind components and $T_v$ with a vertical modulation on the topmost levels. For stabilization purpose, the temporal discretization of GEM also uses an off-centering parameter. The quasi-monotone semi-Lagrangian (QMSL) method (Bermejo and Staniforth, 1992) is used operationally to ensure tracer monotonicity for specific humidity and different hydrometeors. Other options are now available in GEM including a mass

conserving monotonic scheme (Sørenson et al., 2013) and a global mass fixer (Bermejo and Conde, 2002). Those approaches have been evaluated using chemical constituents such as ozone (de Grandpré et al., 2016).

## 6.4    ICON

The ICON model employs damping and diffusion operators for numerical stabilization and dynamic closure. The details of this scheme appear in sections 2.4 and 2.5 of Zängl et al. (2015), and are summarized here. For damping, in the corrector

step a fourth-order divergence damping term $F_d(\mathbf{v})$ is applied in order to allow calling the (relatively) computationally expensive diffusion operator (see below) at the physics time steps without incurring numerical stability problems under extreme conditions,

$$F_d(\mathbf{v}) = -f_d \overline{a_c}^2 \nabla \tilde{\nabla} \cdot \left\{ \nabla \left[ \tilde{\nabla} \cdot v + \frac{\Delta}{\Delta z} \left( w - \overline{\overline{w_{cc}}^c}^i \right) \right] \right\} . \qquad (21)$$

$f_d$ typically attains values between $\frac{1}{1000 \Delta t}$ and $\frac{1}{250 \Delta t}$, and $\overline{a_c}$ is the global mean cell area.

ICON also includes Rayleigh damping on $w$ following Klemp et al. (2008), which serves to prevent unphysical reflections of gravity waves at the model top. The Rayleigh damping is restricted to a fixed number of levels below the model top, and the damping coefficient is given by a hyperbolic tangent.

The horizontal diffusion consists of a flow dependent second-order Smagorinsky diffusion of velocity ($F_{D2}(v_n)$) and potential temperature ($F_{D2}(\theta)$) combined with a fourth-order background diffusion of velocity $F_{D4}(v_n)$, defined via

$F_{D2}(v_n) = 4K_h \tilde{\nabla}^2(v_n), \qquad F_{D2}(\theta) = a_c \tilde{\nabla} \cdot \left( K_h \frac{\Delta \theta}{\Delta n} \right), \qquad F_{D4}(v_n) = -k_4 a_e^2 \tilde{\nabla}^2(\tilde{\nabla}^2(v_n)), \qquad (22)$





where $a_c$ and $a_e$ denotes the area associated with the cell and edge under consideration, respectively. An empirically determined offset of $0.75k_4a_e$ is subtracted from $K_h$ in order to avoid excessive diffusion under weakly disturbed conditions.

A fourth-order computational diffusion is also available for vertical wind speed $w$. This filter term is needed at resolutions of O(1 km) or finer because the advection of vertical wind speed has no implicit damping of small-scale structures. This term appears as

$$F_D(w) = -k_w a_c^2 \nabla^2(\nabla^2(w)). \tag{23}$$

## 6.5 OLAM

OLAM requires two types of artificial damping. In the upper layers of the model, vertical velocity and small-scale horizontal divergence are damped in order to attenuate gravity waves and thereby mitigate their reflection off the rigid top boundary of the domain. The damping layer is commonly applied in the uppermost 10 km of the domain, where the model top is 35 or 40 km above sea level. The damping rate is zero at the bottom of the damping layer and increases upward, usually linearly. Throughout the model domain, vertical vorticity is filtered horizontally at the smallest resolvable scale in order to control a spurious computational mode. This vertical vorticity mode is inherent in C-staggered momentum formulations on hexagonal meshes because horizontal velocities are more numerous than twice the number of scalar (mass) values and are thus under-constrained. The vorticity filter is constructed so as to have zero impact on divergence at any scale. Upwinding in the Lax-Wendroff formulation of the advection operator provides sufficient damping that no other type of filtering is required.

## 7 Temporal discretizations

Temporal discretizations are important for capturing the discrete dynamical evolution of the global atmosphere. In the past two decades, a variety of new temporal discretizations have been developed, leaving behind the days when the leapfrog scheme was ubiquitous across models. This diversity is in part because of the demands of non-hydrostatic models: unlike their hydrostatic counterparts, non-hydrostatic atmospheric models that do not use the anelastic (Ogura and Phillips, 1962), quasi-hydrostatic (Orlanski, 1981), pseudo-incompressible (Durran, 1989), or unified approximation (Arakawa and Konor, 2009) must include a mechanism in the temporal discretization for dealing with vertically propagating sound waves. These waves are meteorologically insignificant, but with a vertical grid spacing of 100 meters, a purely explicit temporal discretization would require a time step size on the order of one second or less. In this section we discuss the timestepping schemes that have been employed across the DCMIP suite of models.

### 7.1 Mixed implicit-explicit, forward-backward, semi-implicit and additive Runge-Kutta schemes

Implicit-explicit schemes are a broad category of time integration schemes that divide the terms of the prognostic equations into a set of explicitly integrated terms and implicitly integrated terms. At the very least, terms associated with vertically propagating sound waves are included among the implicit terms. For the remaining terms, there is some freedom in choosing how to integrate terms associated with vertical advection and horizontally propagating sound waves. Semi-implicit schemes



are one such class of schemes that typically incorporate horizontally propagating sound waves into the implicit solve, and so rely on a global Helmholtz-type solve. Additive Runge-Kutta schemes are another mechanism to ensure high-order temporal accuracy, and many such schemes have been described throughout the literature (see, for example, Weller et al. (2013); Ullrich and Jablonowski (2012b)). Several examples of these schemes can be found among the DCMIP models:

FV$^3$ and its predecessors are integrated using a forward-backwards integration for the Lagrangian dynamics. With the exception of the pressure-gradient force, all of the terms in the momentum, energy, and mass equations are expressable as fluxes, and so can be integrated using the explicit forward-in-time algorithm described by Lin and Rood (1997). The horizontal component of the pressure-gradient force is evaluated backwards-in-time using the algorithm of Lin (1997); the non-hydrostatic component of the vertical pressure gradient force is evaluated using a semi-implicit solver. This forward-backward timestep is

referred to as the "acoustic" timestep, although the full solver is advanced on each of these acoustic timesteps. Physics tendencies are applied impulsively at prescribed intervals, consistent with the forward-in-time discretization; the physics timestep is typically much longer than the acoustic timestep.

DYNAMICO uses an additive Runge-Kutta time scheme with two Butcher tableaus, one explicit and one implicit. A Hamiltonian splitting decides which terms of the equations of motion are treated explicitly or implicitly (Dubos and Dubey, in

preparation). As a result the implicit terms couple the vertical acceleration due to the pressure gradient and the adiabatic pressure change due to vertical displacements of fluid parcels. The resulting implicit problem reduces to independent, scalar, purely vertical, nonlinear problems which are solved to machine precision in two Newton iterations involving one tridiagonal solve each. The overall time scheme has a HEVI (horizontally explicit, vertically implicit) structure. Currently the second-order 3-stage scheme ARK(2,3,2) is used (Giraldo et al., 2013).

ICON consists of a two-time-level predictor corrector scheme, which is explicit for all terms except for those describing the vertical propagation of sound waves. No time splitting is used with respect to sound waves, because the ratio between the speed of sound and the maximum wind speed in the mesosphere, which is in part covered by the vertical domain, can be close to one. Instead time splitting is employed to dynamics on the one hand and horizontal diffusion, tracer transport, fast physics on the other hand. Typically a full time step consists of 4 or 5 dynamical sub-steps in which a constant forcing originating from the

slow physics is applied. Mass-consistent transport is achieved by passing time-averaged air-mass fluxes from the dynamical sub-steps to the transport scheme.The details of the predictor corrector scheme, including measures to increase the numerical efficiency and to optimize the accuracy, are described in section 2.4 of Zängl et al. (2015).

MPAS uses a split-explicit formulation (Klemp et al., 2007) consisting of an outer Runge-Kutta loop (typically RK3) and inner acoustic loop. At the beginning of each Runge-Kutta sub-cycle, tendencies are computed for each of the prognostic

variables and stored for the duration of the sub-cycle. Several iterations of an acoustic loop are then performed with a time-step much smaller than require for the Runge-Kutta sub-cycle. Within the acoustic loop, an implicit solve for vertically-integrated sound waves is performed to avoid timestep constraints that may arise from vertically-propagating sound waves.

OLAM uses a unique temporal discretization that combines elements of the Adams-Bashforth (AB2) scheme and a Lax-Wendroff formulation for advected quantities. However, instead of extrapolating all prognostic tendencies forward to the half-

future time level as in AB2, the horizontal momentum components alone (not their tendencies) are extrapolated in time at the





cell boundaries where they reside. The extrapolated momentum provides the time-centered cell-to-cell total mass flux across the grid cell faces that is responsible for advective transport. Advection of all quantities, including all 3 velocity components that are diagnostically reconstructed at scalar cell centers, and advancement in time from the current to the future time level is based on the time- and space-centered Lax-Wendroff formulation. This scheme is horizontally explicit, but a trapezoidal-

implicit formulation is used in the vertical for stable integration of vertically-propagating sound waves. A by-product of the implicit formulation is an implicit time-centered vertical momentum that joins the time-extrapolated horizontal momentum to form a complete set of mass fluxes for advection. The vertical momentum equation is solved first so that the time-centered vertical momentum is available for computing transport of horizontal momentum and all scalar quantities. A time-split scheme is most often used where momentum and potential temperature are updated more frequently than other scalar fields in order to

accommodate horizontally propagating sound waves.

Tempest uses the ARS(2,3,2) scheme described in Ascher et al. (1997), with all horizontal and vertical advection terms treated explicitly and the remaining vertical terms, associated with sound wave propagation, treated implicitly. A number of different ARK schemes have compared and contrasted in this framework, with significant implications for model performance and stability (Gardner et al., under review).

## 7.2   The FVM Semi-Implicit method

A characteristic feature of the FVM (Section 2.5) time-stepping scheme is the 3D implicit treatment of the fast buoyant and acoustic modes, and the slow rotational modes. Therefore, the model time step is identical for all processes and typically selected with regard to the stability of the advective transport scheme—i.e. the time step is continuously adapted according to a given maximum advective Courant number permitted by the MPDATA scheme. A comprehensive discussion of the integration

scheme can be found in Smolarkiewicz et al. (2014, 2016) and Kühnlein and Smolarkiewicz (2017) for dry dynamics, whereas in Kurowski et al. (2014) and Smolarkiewicz et al. (2017) for extension to moist-precipitating dynamics. Here, we provide a short outline of the solution procedure for the compressible Euler equations (10). It employs the two-time-level second-order-accurate template algorithm given as

$$\psi_{\mathbf{i}}^{n+1} = \mathcal{A}_{\mathbf{i}}(\widetilde{\psi}^n, \mathbf{V}^{n+1/2}, (\mathcal{G}\rho_d)^n, (\mathcal{G}\rho_d)^{n+1}) + 0.5\,\Delta t\, R^\psi|_{\mathbf{i}}^{n+1}\,, \quad \widetilde{\psi}^n \equiv \psi^n + 0.5\,\Delta t\, R^\psi|^n\,. \tag{24}$$

where $\psi$ represents the solution variable, $R^\psi$ is the respective rhs, $\mathcal{A}$ symbolises the advective transport operator given by the non-oscillatory finite-volume MPDATA scheme (Smolarkiewicz and Szmelter, 2005; Kühnlein and Smolarkiewicz, 2017), and the spatial mesh vector index $\mathbf{i} \equiv (k, i)$ denotes the position on the hybrid horizontally-unstructured vertically-structured computational mesh.

The solution procedure of the system (10) can then basically be divided into three steps. First, the homogenous mass con-

tinuity equation (10a) is integrated with $\psi \equiv \rho_d$, $\mathbf{V} \equiv \mathbf{v}\mathcal{G}$, and $R^{\rho_d} \equiv 0$ in (24). Second, given already updated moisture variables (Smolarkiewicz et al., 2017), the thermodynamic (10c) and momentum (10b) equations enter (24) with $\psi = u, v, w, \theta'$, $\mathbf{V} \equiv \mathbf{v}\mathcal{G}\rho_d$, and the rhs $R^\psi$ which is generally depending on all these prognostic variables. The high degree of implicitness in the representation of the rhs forcings is achieved by inverting the overall discrete system (24) to obtain closed-form expressions





for the velocity updates—this procedure is facilitated by the co-located arrangement of all variables on the computational mesh. Retained on the rhs of the derived closed-form velocity expressions is the pressure gradient term. The subsequent third step in the solution procedure is to formulate an implicit boundary value problem for the pressure variable $\phi'$ using an advective form of the equation of state (10d). An $\mathcal{O}(\Delta t^2)$ integration of this equation with a Euler backward scheme, in the spirit of

(24), leads a Helmholtz equation (Smolarkiewicz et al., 2014). The associated 3D elliptic boundary value problem is solved iteratively using a preconditioned Generalised Conjugate Residual approach, see Smolarkiewicz and Szmelter (2011) for a recent overview and comprehensive list of references. Nonlinear terms in $R^\psi|^{n+1}$ and the solution-dependent coefficients of the Helmholtz equation are lagged behind and executed in an outer iteration.

### 7.3    A semi-Lagrangian implicit time discretization in the GEM model

GEM differs from the approaches above by using a semi-Lagrangian advection. Any model equations, prognostic or diagnostic, are written in the form

$$\frac{dF}{dt} + G = 0, \tag{25}$$

where $d/dt$ is the Lagrangian derivative, $F$ containing the terms subject to this operator, $G$ the remaining terms. The semi-Lagrangian approach consists in the following space-time discretization of (25)

$$\frac{F^A - F^D}{\Delta t} + \left(\frac{1}{2} + \epsilon\right) G^A + \left(\frac{1}{2} - \epsilon\right) G^D = 0, \tag{26}$$

where $A$ stands for the arrival position at model grid point $(\mathbf{r}_h, \zeta, t)$ and $D$ for the departure position $(\mathbf{r}_h - \Delta\mathbf{r}_h, \zeta - \Delta\zeta, t - \Delta t)$ due to the displacements $\Delta\mathbf{r}_h, \Delta\zeta$ having occurred during the timestep $\Delta t$. $G$ is averaged between these two positions with a possible slight off-centering $\epsilon$. The displacements are themselves calculated solving, again using the Lagrangian method, the equations:

$$\frac{d\mathbf{r}_h}{dt} - \mathbf{u}_h = 0; \frac{d\zeta}{dt} - \dot\zeta = 0, \tag{27}$$

discretized in the same way (trapezoidal method) though without off-centering:

$$\frac{\Delta\mathbf{r}_h}{\Delta t} - \frac{\mathbf{u}_h{}^A + \mathbf{u}_h{}^D}{2} = 0; \frac{\Delta\zeta}{\Delta t} - \frac{\dot\zeta^A + \dot\zeta^D}{2} = 0. \tag{28}$$

The process is of course a *multi-step iterative* one since both positions and velocities at departure positions (past time $t - \Delta t$) are unknown as well as, of course, the velocities at arrival positions (time $t$). Once a first estimate of the departure positions

is obtained, the model equations are solved to obtain a first estimate of the velocities at time $t$. The model equations must be solved simultaneously and this is only possible for the linear part $L$ which becomes a *matrix inversion problem*. Hence a suitable linearization is considered. The unknown (arrival) linear $L$ and non-linear $N$ parts are then separated from the known (first departure estimate) remaining $R$ part. Thus, first separating space-times, (26) is rewritten as follows

$$\frac{F^A}{\tau} + G^A = \frac{F^D}{\tau} - \beta G^D \equiv R^D, \tag{29}$$



where $\tau = \left(1/2 + \epsilon\right)\Delta t$ and $\beta = \left(1/2 - \epsilon\right)/\left(1/2 + \epsilon\right)$. Second separating linear from non-linear parts, we get:

$$L^A + N^A = R^D, \tag{30}$$

with

$$L^A = \left[\frac{F^A}{\tau} + G^A\right]_{linear}, \quad and \quad N^A \equiv \frac{F^A}{\tau} + G^A - \left[\frac{F^A}{\tau} + G^A\right]_{linear}. \tag{31}$$

Note that both $F$ and $G$ may require linearization. $L^A$ may then be obtained if $N^A$ is first guessed: Once $L^A$ is found, an estimate of $N^A$ is obtained and $L^A$ is recalculated. This is called the *non-linear iteration process* (one iteration is usually sufficient). The overall process is then repeated once starting from a new estimate of the departure positions.

There are two intensive calculation sections in this process: the so-called semi-Lagrangian calculations (twice estimating departure positions, twice interpolating right-hand sides $R$ on departure positions), and solution of the linear system (four times). Each time, the linear system is reduced to a Helmholtz problem for one composite variable. For this problem, a direct solver is involved, using the Schwarz-type domain decomposition method on a Yin-Yang grid (Qaddouri et al., 2008). The composite variable solution is then used to update the prognostic variables (back substitution). At the end of the time-step, the static halo region of both panels of the Yin-Yang grid is updated (Qaddouri and Lee, 2011). All required interpolations throughout the semi-Lagrangian process and between Yin and Yang grids are cubic interpolations.

# 8 Summary and conclusions

As discussed earlier, this paper represents the first in a series of papers documenting the results from the 2016 Dynamical Core Model Intercomparison Project workshop and summer school. In this paper we have provided a description of the differences and similarities between participating models, including the choice of computational grid, horizontal staggering, vertical staggering, vertical coordinates, prognostic equations, choice of diffusion, stabilization, filters and fixers, and temporal discretization. The literature on dynamical core development is vast, with origins that go back over half a century. Consequently, the models discussed in this paper only represent a sample of the many dynamical cores that have been developed for general circulation modeling. Some of the models that have not been discussed include Fox-Rabinovitz et al. (1997); Prusa et al. (2008); Nair et al. (2009); Baba et al. (2010); Donner et al. (2011); Ullrich and Jablonowski (2012a); Gassmann (2013); Wood et al. (2014) and Doyle (2014).

The vast diversity of the modern dynamical core ecosystem suggests that there is no consensus on a single approach that is intrinsically superior to other options. Choices made in the dynamical core confer advantages that include parallel scalability (Dennis et al., 2012), conservation of invariants (Thuburn, 2008), or representation of the kinetic energy spectrum (Skamarock, 2004). The repercussions that emerge from these choices can then be explored in the context of idealized test cases, such as the ones that have been proposed as part of DCMIP. The remaining papers in this series investigate how the models described in this paper are able to simulate three idealized test cases, which each incorporate simplified model physics: a moist baroclinic wave, an idealized tropical cyclone, and a splitting supercell storm on a small planet. Where appropriate, metrics have been




included that may be indicative of model performance. These tests can also be used for future dynamical core development to identify where a new dynamical core diverges from a suite of modern models.

**Code availability**

Information on the availability of source code for the models featured in this paper is tabulated below.

| Short Name | Code availability |
|---|---|
| ACME–A | ACME, including ACME–A, is under active development funded by the U.S. Department of Energy. ACME version 1.0 is scheduled to be publicly released under an open source license in 2018, but is not available at present.[†] |
| CSU | CSU model source code is available under the BSD 3-clause license. The release used for DCMIP2016 can be found via Zenodo (http://dx.doi.org/10.5281/zenodo.580099). |
| DYNAMICO | DYNAMICO is open source and available online from IPSL Forge (http://forge.ipsl.jussieu.fr/dynamico) or directly by request to Thomas Dubos (dubos@lmd.polytechnique.fr). The release used for DCMIP2016 can be found via Zenodo (http://dx.doi.org/10.5281/zenodo.583718). |
| $FV^3$ | $FV^3$ model source code is available through the GFDL Virtual Lab (https://vlab.ncep.noaa.gov/web/fv3gfs). Access requires users to create a Virtual Lab account. |
| FVM | Model codes developed at ECMWF, including the IFS and FVM, are intellectual property of ECMWF and its member states. Although the FVM code is not publicly available at present, it is expected that FVM will be available in the near future under the OpenIFS license (http://www.ecmwf.int/en/research/projects/openifs). The repo tag for the version of FVM that applies for DCMIP is "v0.1".[†] |
| GEM | Due to licensing requirements, GEM model code is only available by request to Abdessamad Qaddouri (Abdessamad.Qaddouri@canada.ca) or Vivian Lee (Vivian.Lee2@canada.ca). |
| ICON | ICON is freely available to the scientific community for non-commercial research under an institutional license issued by project partners DWD+MPI-M. Because of the restrictions of this license, access to the code is only available by request to Günther Zängl (Guenther.zaengl@dwd.de) or Marco Giorgetta (marco.giorgetta@mpimet.mpg.de). |

[†] In compliance with the GMD editorial requirements, this code has been made available to the topical editor in charge of the manuscript.



| Short Name | Code availability (cont'd) |
|---|---|
| MPAS | Open-source via GitHub (https://github.com/MPAS-Dev/MPAS-Release). The release used for DCMIP2016 can be found via Zenodo (http://dx.doi.org/10.5281/zenodo.583316) |
| NICAM | NICAM source code is available under the BSD 2-clause license via Zenodo (http://dx.doi.org/10.5281/zenodo.580128). Further information on collaborating with the NICAM team can be found at http://nicam.jp/hiki/?Research+Collaborations. |
| OLAM | OLAM is open source and available online via SourceForge (https://sourceforge.net/projects/olam-model/). The release used for DCMIP2016 can be found via Zenodo (http://doi.org/10.5281/zenodo.582308). |
| TEMPEST | Tempest source code is available under the Lesser GNU Public License on GitHub (https://github.com/paullric/tempestmodel). The release used for DCMIP2016 can be found via Zenodo (http://dx.doi.org/10.5281/zenodo.579649). |



## Appendix A:  Moist Non-hydrostatic Equation Sets

In this appendix we provide a detailed derivation of the fluid equations utilized by non-hydrostatic models. The physical constants which are used throughout this document is given in Table 6. The material derivative is used for quantities in the Lagrangian frame (following individual air parcels), and is given by

$$\frac{d}{dt} = \frac{\partial}{\partial t} + \mathbf{u} \cdot \nabla, \tag{A1}$$

where $\mathbf{u}$ denotes the 3D vector velocity. Note that tracer variables $q_i$, including specific humidity $q$, in the absence of sources and sinks satisfy the simple Lagrangian relationship

$$\frac{dq_i}{dt} = 0. \tag{A2}$$

**Table 6.** A list of physical constants used in this document.

| Constant | Description | Value |
| --- | --- | --- |
| $a_{\mathrm{ref}}$ | Radius of the Earth | $6.37122 \times 10^6$ m |
| $\Omega_{\mathrm{ref}}$ | Rotational speed of the Earth | $7.292 \times 10^{-5}$ s$^{-1}$ |
| $g_c$ | Gravitational acceleration | 9.80616 m s$^{-2}$ |
| $p_0$ | Reference pressure | 1000 hPa |
| $c_{pd}$ | Specific heat capacity of dry air at constant pressure | 1004.5 J kg$^{-1}$ K$^{-1}$ |
| $c_{pv}$ | Specific heat capacity of water vapor at constant pressure | 1930.0 J kg$^{-1}$ K$^{-1}$ |
| $c_{vd}$ | Specific heat capacity of dry air at constant volume | 717.5 J kg$^{-1}$ K$^{-1}$ |
| $c_{vv}$ | Specific heat capacity of water vapor at constant volume | 1460.0 J kg$^{-1}$ K$^{-1}$ |
| $R_d$ | Gas constant for dry air | 287.0 J kg$^{-1}$ K$^{-1}$ |
| $R_v$ | Gas constant for water vapor | 461.5 J kg$^{-1}$ K$^{-1}$ |
| $\varepsilon$ | Ratio of $R_d$ to $R_v$ | 0.622 |
| $M_v$ | Constant for virtual temperature conversion | 0.608 |
| $\rho_{water}$ | Reference density of water | 1000 kg m$^{-3}$ |

### A1    Diagnostic relationships

The atmospheric fluid is assumed to be an ideal gas. For moist air, the ideal gas constant $R^*$, specific heat capacity at constant pressure $c_p^*$ and specific heat capacity at constant volume $c_v^*$ are given by

$$R^* = R_d + (R_v - R_d)q, \qquad c_p^* = c_{pd} + (c_{pv} - c_{pd})q, \qquad c_v^* = c_{vd} + (c_{vv} - c_{vd})q. \tag{A3}$$

Note that in many models, $R^*$, $c_p^*$ and $c_v^*$ are approximated by $R_d$, $c_{pd}$ and $c_{vd}$, respectively. Dry air, water vapor and moist air quantities all satisfy the linear relationship $R = c_p - c_v$. For a two-fluid system (dry air plus water vapor), two independent



variables plus the specific humidity $q$ are needed to describe the thermodynamic state of the system. Key thermodynamic variables include dry air density $\rho_d$, moist density $\rho$, pressure $p$, vapor pressure $e$, temperature $T$, virtual temperature $T_v$, Exner pressure $\pi$, potential temperature $\theta$, and virtual potential temperature $\theta_v$. Common ratios $\kappa = R^*/c_p^*$, $\varepsilon = R_d/R_v$, and $\gamma = c_p^*/c_v^*$ are adopted here. Note that as additional water species are added (cloud water, rain water, etc.) additional independent variables are needed to capture the thermodynamic effects of these species, and the "virtual" quantities modified accordingly, for instance through the adoption of density potential temperature $\theta_\rho$.

Relationships between key thermodynamic variables arise from the ideal gas law, along with definitions of Exner pressure, potential temperature and virtual potential temperature

$$p = \rho R_d T_v, \qquad \pi = \left(\frac{p}{p_0}\right)^\kappa, \qquad \theta = T \left(\frac{p_0}{p}\right)^\kappa, \qquad \theta_v = T_v \left(\frac{p_0}{p}\right)^\kappa, \tag{A4}$$

which further give rise to

$$p = \left(\frac{\rho R_d \theta_v}{p_0^\kappa}\right)^\gamma, \qquad \pi = \left(\frac{\rho R_d \theta_v}{p_0}\right)^{R^*/c_v^*}, \qquad \theta = \frac{T}{\pi}, \qquad \theta_v = \frac{T_v}{\pi}. \tag{A5}$$

Note that virtual temperature is typically written as

$$T_v = T \left(1 + \frac{(1-\varepsilon)}{\varepsilon} q\right), \tag{A6}$$

which arises from the relationship

$$T_v = \frac{T}{1 - \frac{e}{p}(1-\varepsilon)}, \tag{A7}$$

upon applying $e/p = q/\varepsilon$.

**A2 Prognostic equations for thermodynamic variables**

Note that, as a consequence of (A2), the following simplifications can be applied:

$$\frac{1}{T_v}\frac{dT_v}{dt} = \frac{1}{T}\frac{dT}{dt}, \qquad \frac{dR^*}{dt} = 0, \qquad \frac{dc_p^*}{dt} = 0, \qquad \frac{dc_v^*}{dt} = 0. \tag{A8}$$

Mass conservation is typically represented through the continuity equation, which can be written in the Lagrangian frame as

$$\frac{d\rho}{dt} = -\rho \nabla \cdot \mathbf{u}, \tag{A9}$$

or equivalently in the Eulerian frame,

$$\frac{\partial \rho}{\partial t} = -\nabla \cdot (\rho \mathbf{u}). \tag{A10}$$

Further prognostic relationships can be derived from the thermodynamic equation, including the diabatic heating rate $J$,

$$\frac{1}{T}\frac{dT}{dt} - \frac{\kappa}{p}\frac{dp}{dt} = \frac{J}{T c_p^*}, \tag{A11}$$





which can be alternatively written as

$$\frac{d\theta}{dt} = \frac{J\theta}{Tc_p^*}, \quad \text{or} \quad \frac{d\theta_v}{dt} = \frac{J\theta_v}{Tc_p^*}. \tag{A12}$$

These equations can then be combined with (A9) to obtain

$$\frac{\partial}{\partial t}(\rho\theta_v) + \nabla \cdot (\rho\theta_v \mathbf{u}) = \frac{J\rho\theta_v}{Tc_p^*}, \tag{A13}$$

or similarly for $\theta$. In conjunction with the material derivative of the ideal gas law,

$$\frac{1}{p}\frac{dp}{dt} = \frac{1}{\rho}\frac{d\rho}{dt} + \frac{1}{T_v}\frac{dT_v}{dt}, \tag{A14}$$

the thermodynamic equation can be written in the form

$$\frac{c_v^*}{R^*T_v}\frac{dT_v}{dt} - \frac{1}{\rho}\frac{d\rho}{dt} = \frac{J}{TR^*}. \tag{A15}$$

Then substituting (A9) gives a prognostic equation for virtual temperature,

$$\frac{c_v^*}{R^*}\frac{dT_v}{dt} + T_v\nabla \cdot \mathbf{u} = \frac{JT_v}{TR^*}. \tag{A16}$$

The prognostic equation for temperature is identical except with $T$ substituted for $T_v$. An analogous equations for pressure can be obtained through a similar procedure,

$$\frac{c_v^*}{c_p^*}\frac{dp}{dt} + p\nabla \cdot \mathbf{u} = \frac{Jp}{Tc_p^*}. \tag{A17}$$

And similarly for Exner pressure,

$$\frac{c_v^*}{R^*}\frac{d\pi}{dt} + \pi\nabla \cdot \mathbf{u} = \frac{J\pi}{Tc_p^*}. \tag{A18}$$

### A3   Momentum Equations

In coordinate-invariant form the prognostic velocity equations may be written in either the Lagrangian or Eulerian frame as

$$\frac{d\mathbf{u}}{dt} = \frac{\partial \mathbf{u}}{\partial t} + \mathbf{u} \cdot \nabla\mathbf{u} = -\frac{1}{\rho}\nabla p - 2\mathbf{\Omega} \times \mathbf{u} - \nabla\Phi, \tag{A19}$$

where $\mathbf{\Omega}$ denotes the planetary vorticity vector and $\Phi$ is the geopotential function. The three terms on the right-hand-side of this
expression correspond to pressure gradient, Coriolis, and gravitational force, respectively. In Eulerian form one must be careful with the treatment of the momentum advection term $\mathbf{u} \cdot \nabla\mathbf{u}$, since in an arbitrary coordinate frame this term will give rise to Christoffel symbols associated with derivatives of the vector basis. Note that it is common to rewrite the pressure gradient force using the relationship

$$-\frac{1}{\rho}\nabla p = -\frac{R_d c_p^*}{R^*}\theta_v \nabla\pi, \tag{A20}$$





which follows from (A4). Note that when condensate loading is included, the virtual potential temperautre in this equation is replaced with $\theta_\rho$. A second form of (A19) emerges on substituting the vector calculus identity

$$\mathbf{u} \cdot \nabla \mathbf{u} = \nabla K + \boldsymbol{\zeta} \times \mathbf{u}, \tag{A21}$$

where $K = \frac{1}{2}(\mathbf{u} \cdot \mathbf{u})$ is the 3D kinetic energy and $\boldsymbol{\zeta} = \nabla \times \mathbf{u}$ is the 3D relative vorticity vector. This gives rise to the 3D vector-invariant form,

$$\frac{\partial \mathbf{u}}{\partial t} = -\frac{1}{\rho}\nabla p - \nabla(K + \Phi) - (\boldsymbol{\zeta} + 2\boldsymbol{\Omega}) \times \mathbf{u}. \tag{A22}$$

Because no gradients of vectors appear in this equation, it avoids derivatives of the coordinate basis that would arise from the momentum transport term $\mathbf{u} \cdot \nabla \mathbf{u}$ in (A19). In conjunction with (A9), (A19) also gives rise to the flux-form momentum equations,

$$\frac{\partial}{\partial t}(\rho \mathbf{u}) = -\nabla \cdot (\mathbf{u} \otimes \mathbf{u} + \mathcal{I}p) - 2\boldsymbol{\Omega} \times (\rho \mathbf{u}) - \rho \nabla \Phi, \tag{A23}$$

where $\mathbf{u} \otimes \mathbf{u}$ denotes the outer product and $\mathcal{I}$ is the identity matrix.

The equations above still provide some flexibility with regards to the choice of $\Phi$ and $\boldsymbol{\Omega}$. For *deep atmosphere* models, one typically chooses

$$\Phi = g_c a^2 \left[ \frac{1}{a} - \frac{1}{a+z} \right], \quad \text{and} \quad \boldsymbol{\Omega} = \Omega(\mathbf{k}\sin\varphi + \mathbf{j}\cos\varphi), \tag{A24}$$

where $g_c$ is gravitational acceleration at the surface, $a$ is the radius of the planet, $\Omega$ is the rotation rate (in s$^{-1}$), $\varphi$ is the latitude, $\mathbf{j}$ is the unit vector oriented in the meridional direction, and $\mathbf{k}$ is the unit vector oriented in the vertical direction. For models that don't utilize a height-based vertical coordinate, the geopotential is generally treated as a prognostic variable, with an evolution equation that emerges from the definition $w = dz/dt$,

$$\frac{d\Phi}{dt} = \frac{a^2 g_c w}{(a+z)^2}. \tag{A25}$$

For *shallow atmosphere* models, the geopotential takes the simpler form

$$\Phi = g_c z, \quad \text{and} \quad \boldsymbol{\Omega} = \Omega\sin\varphi \, \mathbf{k}, \tag{A26}$$

where $z$ is the altitude above the surface. In this case we write $2\boldsymbol{\Omega} = f\mathbf{k}$, where $f = 2\Omega\sin\varphi$ is the Coriolis parameter. The evolution equation for the shallow atmosphere geopotential is then

$$\frac{d\Phi}{dt} = g_c w. \tag{A27}$$

## A4    Orthogonal Formulation

Under the orthogonal formulation, projection of a vector field $\mathbf{b}$ onto its horizontal components is defined via

$$[\mathbf{b}]_z = \mathbf{b} - (\mathbf{b} \cdot \mathbf{k})\mathbf{k}. \tag{A28}$$





When applied to the velocity vector this gives rise to the decomposition

$$\mathbf{u} = \mathbf{u}_h + w\mathbf{k}, \tag{A29}$$

where $\mathbf{k} = \nabla z$ is the unit vector in the vertical direction and $\mathbf{u}_h = [\mathbf{u}]_z$ ($\mathbf{u}_h$ is aligned with surfaces of constant $z$). In the orthogonal formulation, the material derivative expands as

$$\frac{d}{dt} = \frac{\partial}{\partial t} + \mathbf{u}_h \cdot \nabla + w\mathbf{k} \cdot \nabla. \tag{A30}$$

For the special case of the material derivative applied to scalars, this equation can also be written as

$$\frac{d}{dt} = \frac{\partial}{\partial t} + \mathbf{u}_h \cdot \nabla_z + w\frac{\partial}{\partial z}. \tag{A31}$$

where $\nabla_z b = [\nabla b]_z$ denotes the gradient along constant $z$ surfaces. From here, the vector-invariant form velocity equation obtained by taking (A22) $\cdot\mathbf{k}$ expands as

$$\frac{\partial w}{\partial t} = -\frac{1}{\rho}\frac{\partial p}{\partial z} - \frac{\partial}{\partial z}(K + \Phi) - [(\boldsymbol{\zeta} + 2\boldsymbol{\Omega}) \times \mathbf{u}] \cdot \mathbf{k}, \tag{A32}$$

which, from $\mathbf{u}_h = \mathbf{u} - w\mathbf{k}$, then gives rise to

$$\frac{\partial \mathbf{u}_h}{\partial t} = -\frac{1}{\rho}\nabla_z p - \nabla_z(K + \Phi) - [(\boldsymbol{\zeta} + 2\boldsymbol{\Omega}) \times \mathbf{u}]_z. \tag{A33}$$

Due to its association with hydrostatic models, it is common to use the 2D kinetic energy, $K_2 = \frac{1}{2}(\mathbf{u}_h \cdot \mathbf{u}_h)$. Decomposing the momentum transport term into horizontal and vertical components gives

$$\mathbf{u} \cdot \nabla \mathbf{u} = \mathbf{u}_h \cdot \nabla \mathbf{u}_h + (\nabla \times \mathbf{u}_h) \times (w\mathbf{k}) + (\mathbf{u} \cdot \nabla w)\mathbf{k}. \tag{A34}$$

The first term in this expression admits the relationships

$$[\mathbf{u}_h \cdot \nabla \mathbf{u}_h]_z = \nabla_z K_2 + \zeta_h \mathbf{k} \times \mathbf{u}_h, \tag{A35}$$

$$(\mathbf{u}_h \cdot \nabla \mathbf{u}_h) \cdot \mathbf{k} = -\mathbf{u}_h \cdot (\mathbf{u}_h \cdot \nabla \mathbf{k}) = -K_2(\nabla \cdot \mathbf{k}) - \frac{1}{2}[\mathbf{u}_h \cdot (\nabla \times \mathbf{u}_t) + (\nabla \times \mathbf{u}_h) \cdot \mathbf{u}_t] \tag{A36}$$

where $\zeta_h = (\nabla \times \mathbf{u}) \cdot \mathbf{k} = (\nabla \times \mathbf{u}_h) \cdot \mathbf{k}$ is the relative vorticity scalar and $\mathbf{u}_t = \mathbf{k} \times \mathbf{u}_h$. Note that this equation does incorporate metric terms associated with horizontal advection of $\mathbf{k}$ which must be accounted for.

Thus the vertical velocity equation, obtained by taking (A19)$\cdot\mathbf{k}$, is

$$\frac{\partial w}{\partial t} = \mathbf{u}_h \cdot (\mathbf{u}_h \cdot \nabla \mathbf{k}) - w\frac{\partial w}{\partial z} - \mathbf{u}_h \cdot \nabla_z w - \frac{\partial \Phi}{\partial z} - \frac{1}{\rho}\frac{\partial p}{\partial z} - (2\boldsymbol{\Omega} \times \mathbf{u}_h) \cdot \mathbf{k}. \tag{A37}$$

Then subtracting (A37)$\cdot\mathbf{k}$ from (A19) gives

$$\frac{\partial \mathbf{u}_h}{\partial t} = -w(\mathbf{u}_h \cdot \nabla \mathbf{k}) - w\frac{\partial \mathbf{u}_h}{\partial z} - \nabla_z(K_2 + \Phi) - \frac{1}{\rho}\nabla_z p - \zeta_h \mathbf{k} \times \mathbf{u}_h - [2\boldsymbol{\Omega} \times \mathbf{u}]_z. \tag{A38}$$

Note that under the shallow atmosphere approximation, the metric term $\mathbf{u}_h \cdot (\mathbf{u}_h \cdot \nabla \mathbf{k})$ in (A37) is set equal to zero in accordance with Phillips (1966).





## A5 Arbitrary vertical coordinates

The dynamical equations are now formulated in terms of the vertical coordinate $s(t, \mathbf{x}, z)$ with $\partial s/\partial z \neq 0$ everywhere, i.e. following Kasahara (1974) (hereafter K74). Since $\mathbf{x}$ and $t$ are shared between the two coordinate systems, the chain rule can be applied to obtain expressions

$$\frac{\partial}{\partial z} = \frac{\partial s}{\partial z} \frac{\partial}{\partial s}, \qquad \nabla_s = \nabla_z + (\nabla_s z)(\mathbf{k} \cdot \nabla), \qquad \left(\frac{\partial}{\partial t}\right)_s = \frac{\partial}{\partial t} + \left(\frac{\partial z}{\partial t}\right)_s (\mathbf{k} \cdot \nabla), \tag{A39}$$

which correspond to derivatives in the vertical, in the horizontal and in time. This final expression is used to describe the rate of change of a quantity on $s$ surfaces. These operators then yield the useful identities

$$\frac{\partial s}{\partial z} = \left(\frac{\partial z}{\partial s}\right)^{-1}, \qquad \nabla_z s = -\left(\frac{\partial s}{\partial z}\right) \nabla_s z, \qquad \frac{\partial s}{\partial t} = -\frac{\partial s}{\partial z} \left(\frac{\partial z}{\partial t}\right)_s. \tag{A40}$$

From here (A39) also gives rise to

$$\nabla_z = \nabla_s - \frac{\partial s}{\partial z}(\nabla_s z)\frac{\partial}{\partial s}, \tag{A41}$$

which can be used directly to rewrite (A33) or (A38) in terms of derivatives over $s$. Note that the operators $\nabla_z$ and $\nabla_s$ are usually introduced in the context of 2D flows, however the construction described here has the advantage of working seamlessly in a 3D context, while admitting the properties $\mathbf{k} \cdot \nabla_z A = 0$ and $\mathbf{k} \cdot \nabla_s A = 0$ for any scalar field $A$.

From (A39), it can be shown that the 2D divergence on $s$ surfaces (given by K74 eq. (3.17)) is

$$\nabla_s \cdot \mathbf{u}_h = \nabla_z \cdot \mathbf{u}_h + \left(\frac{\partial s}{\partial z}\right)(\nabla_s z) \cdot \left(\frac{\partial \mathbf{u}_h}{\partial s}\right), \tag{A42}$$

and that the 2D curl is given by

$$\nabla_s \times \mathbf{u}_h = \nabla_z \times \mathbf{u}_h + \left(\frac{\partial s}{\partial z}\right)(\nabla_s z) \times \left(\frac{\partial \mathbf{u}_h}{\partial s}\right), \tag{A43}$$

where $\nabla_z \times \mathbf{u}_h = \mathbf{k}(\mathbf{k} \cdot (\nabla \times \mathbf{u}_h))$. Notably, these expressions are valid for both shallow- and deep-atmosphere formulations.

The generalized velocity $\dot{s}$ following a fluid parcel is defined by

$$\dot{s} \equiv \frac{ds}{dt} = \frac{\partial s}{\partial t} + \mathbf{u} \cdot \nabla s = \mathbf{u}_h \cdot \nabla_z s + \left[w - \left(\frac{\partial z}{\partial t}\right)_s\right]\frac{\partial s}{\partial z}. \tag{A44}$$

Then using (A39) and (A44) to rewrite (A31) gives an expression for the material derivative for scalars on $s$ surfaces,

$$\frac{dA}{dt} = \left(\frac{\partial A}{\partial t}\right)_s - \frac{\partial s}{\partial z}\left(\frac{\partial z}{\partial t}\right)_s \frac{\partial A}{\partial s} + \mathbf{u}_h \cdot \left[\nabla_s A + (\nabla_z s)\frac{\partial A}{\partial s}\right] + w\frac{\partial s}{\partial z}\frac{\partial A}{\partial s} \tag{A45}$$

$$= \left(\frac{\partial A}{\partial t}\right)_s + \mathbf{u}_h \cdot \nabla_s A + \dot{s}\frac{\partial A}{\partial s} \tag{A46}$$

A similar expression arises for vectors, although in this case $\mathbf{u}_h \cdot \nabla \mathbf{a} \neq \mathbf{u}_h \cdot \nabla_z \mathbf{a}$ implies we cannot use the operator $\nabla_s$ in the form (A39), and instead obtain

$$\frac{d\mathbf{a}}{dt} = \left(\frac{\partial \mathbf{a}}{\partial t}\right)_s + \left[\mathbf{u}_h \cdot \nabla \mathbf{a} + (\mathbf{u}_h \cdot \nabla_s z)(\mathbf{k} \cdot \nabla \mathbf{a})\right] + \dot{s}\frac{\partial \mathbf{a}}{\partial s}. \tag{A47}$$





## A6 Conservation Laws in Arbitrary Vertical Coordinates

Using (A42), we observe that the 3D divergence on the sphere takes the form

$$\nabla \cdot \mathbf{u} = \nabla_z \cdot \mathbf{u}_h + \frac{1}{\alpha}\frac{\partial}{\partial z}(\alpha w), \tag{A48}$$

where $\alpha = 1$ for shallow-atmosphere models and $\alpha = r^2 = (a+z)^2$ for deep-atmosphere models. Using $w = dz/dt$, this last term also takes the form

$$\frac{1}{\alpha}\frac{\partial}{\partial z}(\alpha w) = \frac{\partial w}{\partial z} + \frac{w}{\alpha}\frac{\partial \alpha}{\partial z} = \frac{\partial w}{\partial z} + \frac{1}{\alpha}\frac{d\alpha}{dt}. \tag{A49}$$

Using (A40) to rewrite (A44) gives rise to

$$w = \left(\frac{\partial z}{\partial t}\right)_s + \mathbf{u}_h \cdot \nabla_s z + \dot{s}\left(\frac{\partial s}{\partial z}\right)^{-1}, \tag{A50}$$

which is then differentiated to yield K74 eq. (3.16),

$$\frac{\partial w}{\partial z} = \left(\frac{\partial s}{\partial z}\right)\left[\frac{d}{dt}\left(\frac{\partial s}{\partial z}\right)^{-1} + \left(\frac{\partial \mathbf{u}_h}{\partial s}\right)\cdot(\nabla_s z)\right] + \frac{\partial \dot{s}}{\partial s} = 0. \tag{A51}$$

Substituting this expression into the continuity equation (A9), and using (A48), (A49), and (A51) then leads to

$$\frac{d}{dt}\left[\alpha\left(\frac{\partial s}{\partial z}\right)^{-1}\rho\right] + \alpha\left(\frac{\partial s}{\partial z}\right)^{-1}\rho\left[\nabla_z \cdot \mathbf{u}_h + \left(\frac{\partial s}{\partial z}\right)\left(\frac{\partial \mathbf{u}_h}{\partial s}\right)\cdot(\nabla_s z)\right] + \alpha\left(\frac{\partial s}{\partial z}\right)^{-1}\rho\frac{\partial \dot{s}}{\partial s} = 0. \tag{A52}$$

Defining the *pseudodensity* as

$$\rho_s = \alpha\left(\frac{\partial s}{\partial z}\right)^{-1}\rho, \tag{A53}$$

and using (A46) in the form

$$\frac{d\rho_s}{dt} = \left(\frac{\partial \rho_s}{\partial t}\right)_s + \mathbf{u}_h \cdot \nabla_s \rho_s + \dot{s}\frac{\partial \rho_s}{\partial s}, \tag{A54}$$

along with (A42) leads to

$$\left(\frac{\partial \rho_s}{\partial t}\right)_s + \nabla_s \cdot (\rho_s \mathbf{u}_h) + \frac{\partial}{\partial s}(\rho_s \dot{s}) = 0. \tag{A55}$$

Hence for any quantity that is conserved following a fluid parcel (*i.e.*, $dq/dt = 0$),

$$\left(\frac{\partial \rho_s q}{\partial t}\right)_s + \nabla_s \cdot (\rho_s q \mathbf{u}_h) + \frac{\partial}{\partial s}(\rho_s q \dot{s}) = 0. \tag{A56}$$

In particular, the prognostic equation for virtual potential temperature (or equivalently for potential temperature) reads

$$\left(\frac{\partial \rho_s \theta_v}{\partial t}\right)_s + \nabla_s \cdot (\rho_s \theta_v \mathbf{u}_h) + \frac{\partial}{\partial s}(\rho_s \theta_v \dot{s}) = \frac{J\rho_s \theta_v}{c_p^*}. \tag{A57}$$



## A7    2D Vector Invariant Form

The prognostic equations utilizing horizontal kinetic energy $K_2$ in place of $K$ are derived by applying (A39) to (A37), yielding

$$\left(\frac{\partial w}{\partial t}\right)_s = \mathbf{u}_h \cdot (\mathbf{u}_h \cdot \nabla \mathbf{k}) - \mathbf{u}_h \cdot \nabla w + \left(\frac{\partial s}{\partial z}\right) \left\{ \left[ \left(\frac{\partial z}{\partial t}\right)_s - w \right] \frac{\partial w}{\partial s} - \frac{\partial \Phi}{\partial s} - \frac{1}{\rho}\frac{\partial p}{\partial s} \right\} - (2\boldsymbol{\Omega} \times \mathbf{u}_h) \cdot \mathbf{k}. \tag{A58}$$

Similarly, from (A38),

$$\left(\frac{\partial \mathbf{u}_h}{\partial t}\right)_s = -w(\mathbf{u}_h \cdot \nabla \mathbf{k}) + \left[ \left(\frac{\partial z}{\partial t}\right)_s - w \right] \left(\frac{\partial s}{\partial z}\right) \frac{\partial \mathbf{u}_h}{\partial s} - \zeta_h \mathbf{k} \times \mathbf{u}_h - \nabla_z(K_2 + \Phi) - \frac{1}{\rho}\nabla_z p - [2\boldsymbol{\Omega} \times \mathbf{u}]_z. \tag{A59}$$

Observe that both of these equations simplify when $w = (\partial z/\partial t)_s$, *i.e.* model levels are advected with the vertical wind.

An alternative form of these equation can similarly be obtained in terms of $\dot{s}$. Substituting (A44) into (A58) then gives

$$\left(\frac{\partial w}{\partial t}\right)_s = \mathbf{u}_h \cdot (\mathbf{u}_h \cdot \nabla \mathbf{k}) - \mathbf{u}_h \cdot \nabla_s w - \dot{s}\frac{\partial w}{\partial s} + \left(\frac{\partial s}{\partial z}\right) \left[ -\frac{\partial \Phi}{\partial s} - \frac{1}{\rho}\frac{\partial p}{\partial s} \right] - (2\boldsymbol{\Omega} \times \mathbf{u}_h) \cdot \mathbf{k}. \tag{A60}$$

Similarly, substituting (A44) into (A59) and using the identity

$$(\mathbf{u}_h \cdot \nabla_s z)\left(\frac{\partial s}{\partial z}\right)\frac{\partial \mathbf{u}_h}{\partial s} = (\nabla_s \times \mathbf{u}_h) \times \mathbf{u}_h - \zeta_h \mathbf{k} \times \mathbf{u}_h + (\nabla_s z)\frac{\partial K_2}{\partial z} \tag{A61}$$

then gives

$$\left(\frac{\partial \mathbf{u}_h}{\partial t}\right)_s = -w(\mathbf{u}_h \cdot \nabla \mathbf{k}) - \nabla_s K_2 - \zeta_s \mathbf{k} \times \mathbf{u}_h - \dot{s}\frac{\partial \mathbf{u}_h}{\partial s} - \nabla_z \Phi - \frac{1}{\rho}\nabla_z p - [2\boldsymbol{\Omega} \times \mathbf{u}]_z, \tag{A62}$$

where

$$\nabla_s \times \mathbf{u}_h = \mathbf{k}\zeta_s, \quad \text{and} \quad \zeta_s = \mathbf{k} \cdot (\nabla_s \times \mathbf{u}_h). \tag{A63}$$

In this case the vertical advection terms are removed when $\dot{s} = 0$, *i.e.* the vertical coordinate is advected with the 3D wind $\mathbf{u}$.

Note that under the shallow-atmosphere approximation, the first metric terms (those that include $(\mathbf{u}_h \cdot \nabla \mathbf{k})$) in (A58)-(A62) are typically dropped.

## A8    3D Vector Invariant Form

From (A22) and (A39) the evolution equation for the 3D velocity vector takes the form

$$\left(\frac{\partial \mathbf{u}}{\partial t}\right)_s = \left(\frac{\partial z}{\partial t}\right)_s (\mathbf{k} \cdot \nabla \mathbf{u}) - \nabla(K + \Phi) - \frac{1}{\rho}\nabla p - (\boldsymbol{\zeta} + 2\boldsymbol{\Omega}) \times \mathbf{u} \tag{A64}$$

Then taking the dot product of this expression with $\mathbf{k}$ gives

$$\left(\frac{\partial w}{\partial t}\right)_s = \left(\frac{\partial s}{\partial z}\right) \left[ \left(\frac{\partial z}{\partial t}\right)_s \frac{\partial w}{\partial s} - \frac{\partial}{\partial s}(K + \Phi) - \frac{1}{\rho}\frac{\partial p}{\partial s} \right] - [(\boldsymbol{\zeta} + 2\boldsymbol{\Omega}) \times \mathbf{u}] \cdot \mathbf{k} \tag{A65}$$

where we have used $\mathbf{k} \cdot (\mathbf{k} \cdot \nabla \mathbf{u}) = \mathbf{k} \cdot \nabla w$. Similarly, the prognostic equation for horizontal velocity from (A33) is reformulated as

$$\left(\frac{\partial \mathbf{u}_h}{\partial t}\right)_s = \left(\frac{\partial z}{\partial t}\right)_s \left(\frac{\partial s}{\partial z}\right)\frac{\partial \mathbf{u}_h}{\partial s} - \nabla_z(K + \Phi) - \frac{1}{\rho}\nabla_z p - [(\boldsymbol{\zeta} + 2\boldsymbol{\Omega}) \times \mathbf{u}]_z. \tag{A66}$$





Note that the vorticity term in this expression can be simplified further using

$$[(\boldsymbol{\zeta} + 2\boldsymbol{\Omega}) \times \mathbf{u}]_z = -(\zeta_h + \mathbf{k} \cdot 2\boldsymbol{\Omega})(\mathbf{u}_h \times \mathbf{k}) - w\mathbf{k} \times (\boldsymbol{\zeta} + 2\boldsymbol{\Omega}), \tag{A67}$$

and

$$-\mathbf{k} \times \boldsymbol{\zeta} = \mathbf{k} \cdot \nabla \mathbf{u} - \nabla(\mathbf{k} \cdot \mathbf{u}) + \mathbf{u} \cdot \nabla \mathbf{k} = \frac{\partial \mathbf{u}_h}{\partial z} - \nabla_z w + \mathbf{u}_h \cdot \nabla \mathbf{k}. \tag{A68}$$

## A9  Covariant Component Formulation

In conjunction with (A41), the horizontal momentum equation (in 2D vector invariant form as (A59) or (A62), or in 3D vector invariant form as (A66)) with an arbitrary vertical coordinate gives rise to a two-term pressure gradient. This can be avoided by prognosing the covariant components of the velocity in place of the physical velocity components. We define a horizontal covariance operator by

$$[\mathbf{b}]_s \equiv [\mathbf{b}]_z + (\nabla_s z)(\mathbf{k} \cdot \mathbf{b}). \tag{A69}$$

Applying this operator to the horizontal velocity gives

$$\mathbf{v}_h \equiv [\mathbf{u}]_s = \mathbf{u}_h + (\nabla_s z)w. \tag{A70}$$

For a time-dependent $s$ coordinate, we obtain the identity

$$\left[ \frac{\partial}{\partial t}(\nabla_s z) \right]_s = \nabla_s \left( \frac{\partial z}{\partial t} \right)_s, \tag{A71}$$

and so can write

$$\left( \frac{\partial \mathbf{v}_h}{\partial t} \right)_s = \left( \frac{\partial \mathbf{u}_h}{\partial t} \right)_s + (\nabla_s z)\left( \frac{\partial w}{\partial t} \right)_s + w\nabla_s\left( \frac{\partial z}{\partial t} \right)_s. \tag{A72}$$

Then using (A72), (A65) and (A66) and identity

$$\left( \frac{\partial z}{\partial t} \right)_s \left( \frac{\partial s}{\partial z} \right) \left[ \frac{\partial \mathbf{u}_h}{\partial s} + (\nabla_s z)\frac{\partial w}{\partial s} \right] + w\nabla_s\left( \frac{\partial z}{\partial t} \right)_s$$
$$= \left( \frac{\partial s}{\partial z} \right)\left( \frac{\partial z}{\partial t} \right)_s \left\{ \frac{\partial \mathbf{v}_h}{\partial s} - \nabla_s\left[ \left( \frac{\partial s}{\partial z} \right)^{-1} w \right] \right\} + \nabla_s\left[ \left( \frac{\partial z}{\partial t} \right)_s w \right] \tag{A73}$$

gives

$$\left( \frac{\partial \mathbf{v}_h}{\partial t} \right)_s = -\nabla_s\left[ K - w\left( \frac{\partial z}{\partial t} \right)_s + \Phi \right] - \frac{1}{\rho}\nabla_s p - [(\boldsymbol{\zeta} + 2\boldsymbol{\Omega}) \times \mathbf{u}]_s \tag{A74}$$

$$+ \left( \frac{\partial s}{\partial z} \right)\left( \frac{\partial z}{\partial t} \right)_s \left\{ \frac{\partial \mathbf{v}_h}{\partial s} - \nabla_s\left[ \left( \frac{\partial s}{\partial z} \right)^{-1} w \right] \right\}. \tag{A75}$$

Finally, we can expand the vorticity term and hence obtain

$$\left( \frac{\partial \mathbf{v}_h}{\partial t} \right)_s = -\nabla_s\left[ K - w\left( \frac{\partial z}{\partial t} \right)_s + \Phi \right] - \frac{1}{\rho}\nabla_s p - [2\boldsymbol{\Omega} \times \mathbf{u}]_s$$

$$+ [\mathbf{k} \cdot \nabla_s \times \mathbf{v}_h](\mathbf{u}_h \times \mathbf{k}) - \dot{s}\left\{ \frac{\partial \mathbf{v}_h}{\partial s} - \nabla_s\left[ \left( \frac{\partial s}{\partial z} \right)^{-1} w \right] \right\} - \left( \frac{\partial z}{\partial t} \right)_s \mathbf{u}_h \cdot \nabla \mathbf{k}. \tag{A76}$$





## A10 Vorticity-Divergence Form

The vorticity-divergence form of the dynamical equations in an arbitrary vertical coordinate predicts the absolute vorticity ($\zeta_h^*$) and velocity divergence ($D$) given by

$$\zeta_h^* = (\nabla_s \times \mathbf{u}_h + 2\mathbf{\Omega}) \cdot \mathbf{k}, \tag{A77}$$

and

$$D \equiv \nabla_s \cdot \mathbf{u}_h, \tag{A78}$$

respectively, instead of the horizontal velocity. The horizontal velocity can be obtained from the streamfunction $\psi$ and the velocity potential $\chi$ following

$$\mathbf{u}_h = \mathbf{k} \times \nabla_s \psi + \nabla_s \chi. \tag{A79}$$

By using (A79) in (A77) and (A78), we obtain the elliptic equations that diagnose the streamfunction and velocity potential from the predicted velocity and divergence as

$$\nabla_s^2 = \zeta_h^* - 2\mathbf{\Omega} \cdot \mathbf{k}, \quad \text{and} \quad \nabla_s^2 \chi = D, \tag{A80}$$

respectively.

By taking the material derivative (A46) of (A77) and using the horizontal momentum equation (A38), (A79) and (A80), the

15 absolute vorticity prediction equation emerges,

$$\left(\frac{\partial \zeta_h^*}{\partial t}\right)_s - J_s(\zeta_h^*, \psi) + \nabla_s \cdot (\zeta_h^* \nabla_s \chi) + \nabla_s \cdot \left(\dot{s}\frac{\partial}{\partial s}\nabla_s \psi\right) + \mathbf{k} \cdot \nabla_s \times \left(\dot{s}\frac{\partial}{\partial s}\nabla_s \chi\right) + J_s(\rho^{-1}, p) = 0, \tag{A81}$$

where $J_s(a,b) = \mathbf{k} \cdot \nabla_s \times (a\nabla_s b)$ is the Jacobian operator. It can also be shown that $\dot{s}$ relates to the vertical velocity $w$ through

$$\dot{s} = \left(\frac{\partial s}{\partial z}\right)(w - w_c), \tag{A82}$$

where

$$w_c \equiv \left(\frac{\partial z}{\partial t}\right)_s + (\mathbf{k} \times \nabla_s \psi + \nabla_s \chi) \cdot (\nabla_s z). \tag{A83}$$

By taking the material derivative of (A78) and using (A38), (A79) and (A80), we can obtain the divergence prediction equation

$$\left(\frac{\partial D}{\partial t}\right)_s - J_s(\zeta_h^*, \chi) - \nabla_s \cdot (\zeta_h^* \nabla_s \psi) + \nabla_s \cdot \left(\dot{s}\frac{\partial}{\partial s}\nabla_s \chi\right) + \left(\mathbf{k} \times \frac{\partial}{\partial s}\nabla_s \psi\right) \cdot \nabla_s \dot{s} \tag{A84}$$

$$+ \nabla_s \cdot (\nabla_s K_2 + g\nabla_s z) + \nabla_s \cdot \left(\frac{1}{\rho}\nabla_s p\right) = 0, \tag{A85}$$

where $K_2$ can be reformulated in terms of streamfunction and velocity potential as

$$K_2 = \frac{1}{2}\left[\nabla_s \cdot (\psi \nabla_s \psi) - \psi \nabla_s^2 \psi + \nabla_s \cdot (\chi \nabla_s \chi) - \chi \nabla_s^2 \chi\right] + J_s(\psi, \chi). \tag{A86}$$





### A11   Momentum Form

The momentum form of the prognostic equations emerges by combining the prognostic velocity equations with a continuity equation. Essentially any of the continuity equations can be chosen, as long as the mass field represented by the equation is everywhere non-zero. However, the most common options are moist pseudo-density (Ullrich and Jablonowski, 2012a) or dry pseudo-density (Skamarock et al., 2012). Here we denote our density variable by $\tilde{\rho}_s$, and assume no external sources or sinks of $\tilde{\rho}$. Multiplying (A60) through by $\tilde{\rho}_s$ and using (A55) gives

$$\left(\frac{\partial \tilde{\rho}_s w}{\partial t}\right)_s = \tilde{\rho}_s \mathbf{u}_h \cdot (\mathbf{u}_h \cdot \nabla \mathbf{k}) - \nabla_s \cdot (\tilde{\rho}_s \mathbf{u}_h w) - \frac{\partial}{\partial s}(\tilde{\rho}_s \dot{s} w) + \tilde{\rho}_s \left(\frac{\partial s}{\partial z}\right) \left[-\frac{\partial \Phi}{\partial s} - \frac{1}{\rho}\frac{\partial p}{\partial s}\right] - (2\mathbf{\Omega} \times \tilde{\rho}_s \mathbf{u}_h) \cdot \mathbf{k}. \tag{A87}$$

Similarly, from (A62) we have

$$\left(\frac{\partial \tilde{\rho}_s \mathbf{u}_h}{\partial t}\right)_s = -\tilde{\rho}_s w(\mathbf{u}_h \cdot \nabla \mathbf{k}) - \tilde{\rho}_s \nabla_s K_2 - \zeta_s \mathbf{k} \times \tilde{\rho}_s \mathbf{u}_h - \mathbf{u}_h \nabla_s \cdot (\tilde{\rho}_s \mathbf{u}_h) - \frac{\partial}{\partial s}(\tilde{\rho}_s \dot{s} \mathbf{u}_h) - \tilde{\rho}_s \left(\nabla_z \Phi + \frac{1}{\rho}\nabla_z p\right) - [2\mathbf{\Omega} \times \tilde{\rho}_s \mathbf{u}]_z, \tag{A88}$$

*Author contributions.*   Text in this manuscript describing individual models was provided by the respective modeling teams. Final composition and development of Appendix A was performed by P.A. Ullrich.

*Acknowledgements.*   DCMIP2016 is sponsored by the National Center for Atmospheric Research Computational Information Systems Laboratory, the Department of Energy Office of Science (award no. DE-SC0016015), the National Science Foundation (award no. 1629819), the National Aeronautics and Space Administration (award no. NNX16AK51G), the National Oceanic and Atmospheric Administration Great Lakes Environmental Research Laboratory (award no. NA12OAR4320071), the Office of Naval Research and CU Boulder Research Computing. This work was made possible with support from our student and postdoctoral participants: Sabina Abba Omar, Scott Bachman, Amanda Back, Tobias Bauer, Vinicius Capistrano, Spencer Clark, Ross Dixon, Christopher Eldred, Robert Fajber, Jared Ferguson, Emily Foshee, Ariane Frassoni, Alexander Goldstein, Jorge Guerra, Chasity Henson, Adam Herrington, Tsung-Lin Hsieh, Dave Lee, Theodore Letcher, Weiwei Li, Laura Mazzaro, Maximo Menchaca, Jonathan Meyer, Farshid Nazari, John O'Brien, Bjarke Tobias Olsen, Hossein Parishani, Charles Pelletier, Thomas Rackow, Kabir Rasouli, Cameron Rencurrel, Koichi Sakaguchi, Gökhan Sever, James Shaw, Konrad Simon, Abhishekh Srivastava, Nicholas Szapiro, Kazushi Takemura, Pushp Raj Tiwari, Chii-Yun Tsai, Richard Urata, Karin van der Wiel, Lei Wang, Eric Wolf, Zheng Wu, Haiyang Yu, Sungduk Yu and Jiawei Zhuang. We would also like to thank Rich Loft, Cecilia Banner, Kathryn Peczkowicz and Rory Kelly (NCAR), Perla Dinger, Carmen Ho, and Gina Skyberg (UC Davis) and Kristi Hansen (University of Michigan) for administrative support during the workshop and summer school.





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
