# Peer review of "DCMIP2016: A Review of Non-hydrostatic Dynamical Core Design and Intercomparison of Participating Models"

_Geoscientific Model Development, 2017_

## Referee Comment (RC1) · H. Weller (Referee) · 27 Jun 2017

This is an exceptionally well written paper leading to the most minor review that I have every written. The paper describes the 11 models which took part in DCMIP 2016 with clarity, consistency and the kind of insight that very few people have. This will be an extremely useful resource for those seeking to understand how any of the models work and how they compare to other models. I have a very few minor comments.

1. OLAM uses cut cells to represent smooth topography - how is the small cell problem solved.

2. Perhaps mention the advantages of the lat-lon grid

3. In section 6.5, give a citation describing the hexagonal C-grid computational mode and either give a citation or describe the filter.

4. In section 7, you say that fully compressible non-hydrostatic models need a temporal discretisation for dealing with vertically propagating sound waves. You give the impression that using some form of approximation that filters sound waves implies that the problem goes away. It doesn't, it makes the problem elliptic rather than hyperbolic and so requires the solution of a Poisson equation rather than a Helmholtz equation. It is a common misconception that these approximations somehow make solving the equations easier. Please help to dispel this misconception.

5. Line 29 of page 28. Remove the word "basically".

---

## Referee Comment (RC2) · T. Melvin (Referee) · 28 Jun 2017

This paper gives an overview of the models that participated in the DCMIP2016 workshop on dynamical core intercomparisons. By itself the paper provides a useful reference on the current state of the art in regards to dynamical core development, highlighting the wide range of choices that have been made by modeling groups across the globe as well as highlighting some of the choices, such as equation sets, used in dynamical core design. As noted by the authors this paper is the first of an envisioned sequence detailing the models and their performance on a number of idealized test cases and it will be interesting to observe how the sequence develops and what can

be learned from the intercomparison.

It is some achievement to condense the wealth of information needed to describe a range of modeling issues into one concise paper and the authors should be applauded for succeeding in such a difficult task. The paper is well written and provides a useful source of information to model developers and is therefore suitable for publication after a number of minor issues are dealt with.

Main Comments

1. The main issue that should be corrected in this paper is that not all of the models are covered in each of sections 3-7 describing aspects of the model formulations. This could be due a desire not to replicate information (if models share the same governing equations etc) but I think it would be useful if the reader could find the appropriate model description in each of sections 3-7. In detail I suggest:

   - Section 5 lists the equation sets used by each model but is missing the CSU, MPAS and NICAM models. It would be useful for the reader to add brief sections for these models, or if they are the same as some of the other models to combine them into the appropriate subsection.

   - Section 6 describes the diffusion mechanisms in each model but omits the CSU, DYNAMICO, FVM, MPAS and NICAM models. Since table 5 indicates these models to have explicit diffusion mechanisms then it would be good to add subsections for the missing models, or where appropriate combine them, e.g. CSU and DYNAMICO both use 4th order hyperviscosity which is covered in subsection 6.1 on ACME-A and so these models could be combined into a single section.

   - Related to the previous point the methods of diffusion & stabilization in table 5 and section 6 are somewhat different, for example some model subsec-

tions describe using sponge layers (FV3, ICON) but these are not listed in table 5 and the same applies to monotonic limiting for some models. Is it the case that table 5 only lists the principle methods of stabilization and diffusion? In which case I suggest adding words to this effect in the caption. I appreciate that it is beyond the scope of this paper to list in detail all the methods of stabilization and diffusion applied in all the models, maybe some words in the introduction to section 6 indicating that this section only covers the principle diffusion methods used?

- Section 6 lists the temporal discretization methods used but omits the methods used by the ACME-A, CSU and NICAM models, it would be useful it the methods used by these models was indicated in this section.

2. Section 2 gives a brief description of each model and as noted in the author contributions these are provided by the modeling teams themselves, however this has led to a rather uneven section where the model descriptions provide differing levels of detail. I think this section could use some editorial input to unify the descriptions. Based upon the sections for the rest of the paper I would like to be able to ascertain the following properties for each model from this section: equation set, horizontal grid and discretization, vertical grid and discretization, temporal discretization, principal diffusion and stabilization mechanisms and transport scheme. Only a couple of words to a sentence are needed and much of this information can be found in tables 2-5 but i think it would help readability to unify this description section.

3. The paper does a very good job of describing the key features of a wide range of models, however I would have been interested in seeing a specific section detailing the transport schemes used by the models in a similar fashion to Sections 3-7 (and including the information in table 2 if possible). However in order to avoid over lengthening the paper I suggest this could be covered to some extent by the descriptions in section 2. This would require details of the transport schemes

used by ACME-A, OLAM and TEMPEST to be mentioned in the appropriate sub-sections of section 2.

Minor Comments

1. Table 1: $\Phi$, $\delta\Phi$, $\dot{\zeta}$ and $\theta'$ are missing entries but listed in the prognostic variables of table 3.

2. The DCMIP2016 website lists HOMME, UZIM and NEPTUNE (NEP) as models taking part, I assume that HOMME is ACME-A, UZIM is CSU, if I'm mistaken then could these models be added?. Is there a reason NEPTUNE is not included in this paper?

3. Section 2.3: If Dubos and Dubery has been submitted this reference could be updated.

4. Section 2.7 'Icosahedral' should be 'ICOsahedral' in the subsection title to match the format of other model names.

5. Section 3.5, last line: Is it possible that the CCVT method produces polygons with less than 5 sides? If so this should be mentioned.

6. Section 3.7 last line. I don't think it is entirely correct that GEM uses two regional climate models on the patches of the YinYang grid. Qaddouri 2011 States that the numerics come from the original GEM latlong model which is used for medium range weather forecasts. I suggest changing this to "utilizing a pair of local area models based with the numerics from the GEM latitude-longitude model". If the GEM modeling team feel the current description is accurate then I am happy for it to be left as is.

7. Page 16, Line 8. The A-grid collocates all scalar and velocity components. To avoid confusion with the B- and E-grid (which only collocate velocity components) I suggest changing "co-location of all velocity components" to "co-location of all velocity components and scalar fields".

8. Page 16 Line 9: To be consistent with the descriptions of the other grids I would add "which co-locates the vorticity, divergence and buoyancy variable." after "and the Z-grid".

9. Page 16 Line 14: There is a mix up of dimensionality of the mesh objects here, for a 3D mesh the C-grid stores velocities on faces not edges. I would suggest saying "as long as the number of horizontal faces is twice the number of volumes".

10. Page 16 Lines 1 and 14-15: The maximum stable timestep size (if it exists) is given by a combination of factors such as the time scheme, horizontal and vertical discretization, grid staggering and waves supported in the equation set. The comments in this section give the reader the impression that staggering is the most important (or only) factor. I suggest that "for explicit timestepping schemes" is added after "timestep size" on line 1 and that the text on lines 14-15 from "but also" to the semi-colon is removed since I believe this statement is only true for a given choice of horizontal discretization (2nd order fd?) and defined explicit timestepping schemes.

11. Page 16 Line 18: In general I think it is a Poisson problem that needs to be solved for the z-grid rather than the more general Helmholtz problem.

12. Page 28 Line 14: Could a citation (at least title and authors) be given for this paper if it is under review.

Typos

1. Page 3 Line 12 "provide" -> "provides"

2. Page 20 Line 16 Missing "are" after "employed"

3. Page 33 Line 3 "is given" -> "are given"

---

## Editor Comment (EC1) · J. C. Hargreaves (Editor) · 9 Sep 2017

I reopened the online peer review a few weeks ago as I had not yet gained access to all of the three not-publicly-accessible codes (ACME, GEM and FVM), and I was also hoping for input from the third reviewer who had agreed to look at the manuscript. I have now been given access to all three codes. This required some effort on the part of the authors, for which I am grateful. This manuscript is not in the model description paper type, and is much closer to being a review article, and I do not see reason to be critical over anything related to the code - I will just say that I enjoyed seeing how the different developers have approached the design of their modelling infrastructure.

[Figure]

Before submitting a revision, I would appreciate it if the lead author of the paper would test the veracity of the links to (or other ways of accessing) the other models included in the code accessibility table. In my experience there are often small issues that need to be resolved to make sure the information is accurate and that the code is really accessible to all.

I have received some comments from a third reviewer, who did not like the paper at all, but declined to give a full review. I will pass on a paraphrased version of their comments as, particularly in relation to the organisation of the paper, I do see where they are coming from. Please consider their complaints seriously, respond to each point, and revise the structure of the paper to make it more logically organised and readable. It is possible that this reviewer did not take the time to consider the nature of a GMD paper. I am encouraged in this thinking by the fact that one of the other reviewers is a former GMD editor, and they gave a very encouraging review. In that context I am still considering this to be a minor revision despite this negative review.

Paraphrased comments from anonymous 3rd reviewer.

The reviewer commented to the effect that. . .

The purpose of the paper is not clear.

There is nothing of interest or importance to the modelling community in the paper.

The paper is too long, has too many equations and all of it is already published in the literature.

The material is presented in a highly confusing manner, "for instance, Section 2 is sub-divided by model (with 11 sections, 1 per model), but Section 3 only has 8 subsections (why not 11?), and Section 4 only 2 subsections, the second of which only has 4 sub-subsection (yet, when you look at Table 4, last column, I only see 3 different methods, so I have no idea where there are 4 sub-subsections for 3 methods); then Section 5 is again subsectioned by model (why this back and forth), but instead of 11 I only see 8

subsections: why are 3 models missing?"

[editor's note: GMD papers are allowed to be long and may contain many equations, but because of this, an easy to navigate logical and readable structure is even more important than it is for regular science papers.]

---

## Author Response (AR1)

**This is an exceptionally well written paper leading to the most minor review that I have every written. The paper describes the 11 models which took part in DCMIP 2016 with clarity, consistency and the kind of insight that very few people have. This will be an extremely useful resource for those seeking to understand how any of the models work and how they compare to other models. I have a very few minor comments.**

We would like to thank the reviewer for this very kind comment. We are very hopeful that this manuscript will be a valuable resource for future model developers and students of model development.

**1. OLAM uses cut cells to represent smooth topography - how is the small cell problem solved.**

To address this question, the following text has been added to section 4.2.4:

One or more methods are used to avoid the so-called small cell problem where volume to area ratios of cut cells are much less than for full cells and therefore can lead to instability. The smallest cells are eliminated by adjusting topography slightly, which is usually justified by noting that local topographic sampling is approximate. In larger cut cells, volumes can be increased (without changing surface areas) which stabilizes the cell at the expense of slowing its response to advected transients. When either of the above adjustments is unacceptable for a particular application, a flux-balance method based partly on Berger and Helzel (2012) is used to stabilize small cut cells.

**2. Perhaps mention the advantages of the lat-lon grid.**

Agreed. We have added the following text to the description of the latitude-longitude grid:

The latitude-longitude grid has the benefits of being globally rectilinear, which simplifies data access and subdivision of computation across processors, and yields a vector basis that is locally orthogonal nearly everywhere. This structure accurately maintains purely zonal flows and simplifies data post-processing for visualization.

**3. In section 6.5, give a citation describing the hexagonal C-grid computational mode and either give a citation or describe the filter.**

Agreed. In the revised manuscript we have included citations to Weller, H.: Controlling the computational modes of the arbitrarily structured C grid, Mon. Weather Rev., 140, 3220–3234, 2012 and Weller, H., Thuburn, J., and Cotter, C. J.: Computational modes and grid imprinting on five quasi-uniform spherical C grids, Mon. Weather Rev., 140, 2734–2755, 2012.

**4. In section 7, you say that fully compressible non-hydrostatic models need a temporal discretisation for dealing with vertically propagating sound waves. You give the impression that using some form of approximation that filters sound waves implies that the problem goes away. It doesn't, it makes the problem elliptic rather than hyperbolic and so requires the solution of a Poisson equation rather than a Helmholtz equation. It is a common misconception that these approximations somehow make solving the equations easier. Please help to dispel this misconception.**

Thank you for pointing this out. The text has been modified to read:

This diversity is in part because of the demands of non-hydrostatic models: unlike their hydrostatic counterparts, non-hydrostatic atmospheric models must include a mechanism for dealing with vertically propagating sound waves. These waves are meteorologically insignificant, but with a vertical grid spacing of 100 meters, a purely explicit temporal discretization of the unmodified fluid equations would require a time step size on the order of one second or less. Consequently, sound waves are either filtered explicitly through the use of an alternative equation set, or artificially slowed through the use of implicit temporal discretizations.

Some commonly employed alternative equation sets include the anelastic (Ogura and Phillips, 1962), quasi-hydrostatic (Orlanski, 1981), pseudo-incompressible (Durran, 1989), or unified approximation (Arakawa and Konor, 2009). These filtered equation sets generally require that a global elliptic solve be performed as prognostic variables are updated.

**5. Line 29 of page 28. Remove the word "basically".**

Removed.

This paper gives an overview of the models that participated in the DCMIP2016 workshop on dynamical core intercomparisons. By itself the paper provides a useful reference on the current state of the art in regards to dynamical core development, highlighting the wide range of choices that have been made by modeling groups across the globe as well as highlighting some of the choices, such as equation sets, used in dynamical core design. As noted by the authors this paper is the first of an envisioned sequence detailing the models and their performance on a number of idealized test cases and it will be interesting to observe how the sequence develops and what can be learned from the intercomparison.

It is some achievement to condense the wealth of information needed to describe a range of modeling issues into one concise paper and the authors should be applauded for succeeding in such a difficult task. The paper is well written and provides a useful source of information to model developers and is therefore suitable for publication after a number of minor issues are dealt with.

We thank the reviewer for his positive comments on the manuscript. We agree that, in the interests of completeness, each section should feature descriptions from all models being assessed and so have added text to this effect to each of sections 3-7. Further, the model descriptions in section 2 have been rewritten to maintain a more consistent structure.

**Main comments:**

**1. The main issue that should be corrected in this paper is that not all of the models are covered in each of sections 3-7 describing aspects of the model formulations. This could be due a desire not to replicate information (if models share the same governing equations etc) but I think it would be useful if the reader could find the appropriate model description in each of sections 3-7. In detail I suggest:**

We agree with your assessment and have added the missing sections, as requested.

**Section 5 lists the equation sets used by each model but is missing the CSU, MPAS and NICAM models. It would be useful for the reader to add brief sections for these models, or if they are the same as some of the other models to combine them into the appropriate subsection.**

Text for CSU, MPAS and NICAM has been added to this section:

**CSU:** The CSU model uses the vorticity-divergence form of the equations of motion, as described in section A10, discretized on the geodesic mesh with absolute vorticity and velocity divergence scalars stored at cell-centers. The unified approximation of the equations of motion (Arakawa and Konor, 2009) is employed to avoid vertically propagating sound waves.

**MPAS:** The evolution equations used by MPAS are fully described in Skamarock et al. (2012), based on the formulation of Dutton (1986). The MPAS model uses the momentum form of the update equations, as described in section A11, with dry mass utilized for the density variable $\tilde{\rho}_s$. MPAS further evolves dry mass using a continuity equation of the form (A10) and moist potential temperature following (A13).

**NICAM:** NICAM prognoses horizontal and vertical momentum analogous to the approach described in section A11. It further evolves the density perturbation from the background reference state using (A10) and sensible heat part of internal energy. A detailed explanation of the evolution equations can be found in Satoh et al. (2008).

**Section 6 describes the diffusion mechanisms in each model but omits the CSU, DYNAMICO, FVM, MPAS and NICAM models. Since table 5 indicates these models to have explicit diffusion mechanisms then it would be good to add subsections for the missing models, or where**

**appropriate combine them, e.g. CSU and DYNAMICO both use 4th order hyperviscosity which is covered in subsection 6.1 on ACME-A and so these models could be combined into a single section.**

The following text has been added in section 6.

**CSU:** The CSU model utilizes an explicit diffusion scheme that consists of fourth-order hyperdiffusion ($\nabla^4$) applied to the vorticity, divergence, and potential temperature. The model does not include any explicit diffusion in the vertical column. However, for the idealized DCMIP test cases explicit diffusion was disabled.

**DYNAMICO:** In DYNAMICO, (hyper-)diffusive filters are used to eliminate spurious noise due to the energy-conservative centered discretization. Filters are applied every $N_{diff}$ Runge-Kutta time steps in a forward-Euler manner, with $N_{diff}$ as large as allowed by stability. The scalar Laplacian is computed as div grad and the vector Laplacian is decomposed into its divergent (grad div) and rotational (curl curl) parts. The strength of filtering is controlled by dissipation time scales $\tau$: Given $\tau$ the hyperviscous coefficient that multiplies operator $D^p$ is $\delta^{2p}\tau^-1$ where $\delta^-2$ is the largest eigenvalue of operator $D$. For DCMIP, DYNAMICO uses $p = 2$ (fourth-order hyperviscosity) for all filters.

**FVM:** Within the dynamical core, FVM does not apply any explicit diffusion. For the DCMIP test cases, the implicit diffusion of the monotonic MPDATA provides the right amount of diffusion/dissipation needed to maintain model stability and remove excess energy from the finest scales. An absorbing layer is also available in the model for damping vertically propagating waves near the model top.

**MPAS:** The MPAS model applies fourth-order hyperdiffusion and Smagorinsky diffusion (Smagorinsky, 1963), as described in Skamarock et al. (2012). When applied to the momentum, the Laplacian is evaluated as

$$\nabla^2 u_i = \frac{\partial}{\partial x_i}\nabla_s \cdot \mathbf{v} - \frac{\partial \eta}{\partial x_j}, \tag{1}$$

where $u_i$ is the edge-normal velocity defined on cell edge $i$, $\eta$ is the vertical component of the relative vorticity, computed on vertices, and $\nabla_s \cdot \mathbf{v}$ is the horizontal divergence on $s$ surfaces, computed on edges. The evaluation of divergence and vorticity in this expression is described in [**?**]. The fourth-order hyperdiffusion operator is then computed by twice applying the above Laplacian operator to the momentum.

Smagorinsky diffusion, which is often applied in atmospheric models to parameterize turbulent processes, uses a second-order Laplacian and a physically-motivated eddy viscosity $K_h$, defined in terms of Cartesian velocities $(u, v)$,

$$K_h = c_s^2 \ell^2 \sqrt{(u_x - v_y)^2 + (u_y + v_x)^2}, \tag{2}$$

where $c_s$ is a constant parameter and $\ell$ is the grid scale. The diffusion operator then takes the form $\nabla \cdot (K_h \nabla \psi)$ for a scalar field $\psi$.

**NICAM:** NICAM implements three types of diffusion: 3D divergence damping, fourth-order horizontal hyperdiffusion, and sixth-order vertical hyperdiffusion as described in Tomita and Satoh (2004). Specifically, the divergence damping term (Skamarock and Klemp, 1992) aims to suppress instabilities that arise due to the time splitting scheme, and is applied to both horizontal and vertical velocities. The hyperdiffusion operators are applied to all prognostic variables. For tracer advection, upwinding is used to remove spurious oscillations, as described in Miura (2007); Niwa et al. (2011).

**Related to the previous point the methods of diffusion & stabilization in table 5 and section 6 are somewhat different, for example some model subsections describe using sponge layers (FV3, ICON) but these are not listed in table 5 and the same applies to monotonic limiting for some models. Is it the case that table 5 only lists the principle methods of stabilization and diffusion? In which case I suggest adding words to this effect in the caption. I appreciate that**

**it is beyond the scope of this paper to list in detail all the methods of stabilization and diffusion applied in all the models, maybe some words in the introduction to section 6 indicating that this section only covers the principle diffusion methods used?**

The header of Table 5 has been changed to read "Principal options for diffusion, stabilization, filters or fixers." Further, the following sentence has been added to the introduction of section 6:

Diffusion also includes mechanisms for damping vertically propagating internal gravity waves, such as model-top Rayleigh layers, which are fairly ubiquitous across models and hence not discussed in detail here.

**Section 6 lists the temporal discretization methods used but omits the methods used by the ACME-A, CSU and NICAM models, it would be useful it the methods used by these models was indicated in this section.**

ACME-A uses the same method as Tempest, so these have been combined in the text. NICAM uses the same method as MPAS, so these have also been combined in the text.

The following text has been added for the CSU model:

CSU uses a semi-implicit time integration scheme with third-order Adams-Bashforth scheme for explicit integration of the continuity equation, potential temperature equation, and terms related to advection. Since potential temperature is updated prior to the computation of the pressure-gradient force, this term can be thought of as implicit in time. The horizontal wind field is then predicted through integration of the vorticity and divergence of the horizontal wind and a multigrid method applied to solve a pair of two-dimensional Poisson equations for the stream function and velocity potential, which are then differentiated to obtain the velocity field. Horizontal diffusion is then applied forward in time.

**2. Section 2 gives a brief description of each model and as noted in the author contributions these are provided by the modeling teams themselves, however this has led to a rather uneven section where the model descriptions provide differing levels of detail. I think this section could use some editorial input to unify the descriptions. Based upon the sections for the rest of the paper I would like to be able to ascertain the following properties for each model from this section: equation set, horizontal grid and discretization, vertical grid and discretization, temporal discretization, principal diffusion and stabilization mechanisms and transport scheme. Only a couple of words to a sentence are needed and much of this information can be found in tables 2-5 but i think it would help readability to unify this description section.**

Thank you for this suggestion. Section 2 has been modified heavily in order to better align the wording for each model. The changes are sufficiently extensive that they are not reproduced here.

**3. The paper does a very good job of describing the key features of a wide range of models, however I would have been interested in seeing a specific section detailing the transport schemes used by the models in a similar fashion to Sections 3-7 (and including the information in table 2 if possible). However in order to avoid over lengthening the paper I suggest this could be covered to some extent by the descriptions in section 2. This would require details of the transport schemes used by ACME-A, OLAM and TEMPEST to be mentioned in the appropriate subsections of section 2.**

The following updates to the text have been made:

**ACME-A:** By default, tracer transport is sub-cycled relative to the hydrodynamics, but also performed using the spectral element method using the tracer mass as a prognostic variable.

**OLAM:** Tracer transport is second-order in space and time using the scheme of Miura (2007), with consistent

fluxes obtained by time averaging over the acoustic time steps.

**TEMPEST:** Tracer transport is performed using the spectral element method with the same timestep as the hydrodynamics and using the tracer mass density as a prognostic variable. As with the hydrodynamics, tracer transport is updated explicitly in the horizontal and implicitly in the vertical.

**Minor comments:**

**Table 1: $\Phi$, $\delta\Phi$, $\dot{\zeta}$ and $\theta'$ are missing entries but listed in the prognostic variables of table 3.**

Thank you for catching this. Entries have been added for $\Phi$ and $\dot{\zeta}$ in Table 1. In Table 3, $\delta\Phi$ has been changed to $\Phi$ and $\theta'$ has been changed to $\theta$.

**The DCMIP2016 website lists HOMME, UZIM and NEPTUNE (NEP) as models taking part, I assume that HOMME is ACME-A, UZIM is CSU, if I'm mistaken then could these models be added? Is there a reason NEPTUNE is not included in this paper?**

HOMME was used at DCMIP2016 using a hydrostatic formulation. This model has since been updated to include a nonhydrostatic formulation that was subsequently included in ACME-A. Since it would be odd to include a model that was purely hydrostatic in a paper that focused on nonhydrostatic models, it was decided that HOMME would be substituted for ACME-A.

You are correct that UZIM and CSU are the same model (although a new curl-curl form model is now under development at CSU).

NEPTUNE participated in the DCMIP2016 workshop but did not submit text for this paper describing their model, and so are not included. Since they did invest the considerable effort in participating in the workshop, we believed it would be appropriate to retain them as co-authors even though the model description was absent.

**Section 2.3: If Dubos and Dubery has been submitted this reference could be updated.**

Although there is not much remaining to be done, this paper has not been submitted as of the time of writing.

**Section 2.7 "Icosahedral" should be "ICOsahedral" in the subsection title to match the format of other model names.**

Corrected.

**Section 3.5, last line: Is it possible that the CCVT method produces polygons with less than 5 sides? If so this should be mentioned.**

Yes, although it's very unlikely. The text has been updated as follows:

Although hexagons are, by far, the most common polygon on CCVT grids, CCVT grids on the sphere will also include at least 12 pentagons and sometimes other polygons with more than six sides. Quadrilateral elements are theoretically possible, but are never found in practice on the final grid due to being a locally unstable solution of the underlying CCVT system of equations.

**Section 3.7 last line. I don?t think it is entirely correct that GEM uses two regional climate models on the patches of the YinYang grid. Qaddouri 2011 States that the numerics come from the original GEM latlong model which is used for medium range weather forecasts. I suggest changing this to "utilizing a pair of local area models based with the numerics from**

the GEM latitude-longitude model". If the GEM modeling team feel the current description is accurate then I am happy for it to be left as is.

We agree with the suggested change. The modified text has been incorporated in the manuscript.

**Page 16, Line 8. The A-grid collocates all scalar and velocity components. To avoid confusion with the B- and E-grid (which only collocate velocity components) I suggest changing "co-location of all velocity components" to "co-location of all velocity components and scalar fields".**

Agreed and changed accordingly.

**Page 16 Line 9: To be consistent with the descriptions of the other grids I would add "which co-locates the vorticity, divergence and buoyancy variable." after "and the Z-grid".**

Agreed and changed accordingly.

**Page 16 Line 14: There is a mix up of dimensionality of the mesh objects here, for a 3D mesh the C-grid stores velocities on faces not edges. I would suggest saying "as long as the number of horizontal faces is twice the number of volumes".**

Agreed and changed accordingly.

**Page 16 Lines 1 and 14-15: The maximum stable timestep size (if it exists) is given by a combination of factors such as the time scheme, horizontal and vertical discretization, grid staggering and waves supported in the equation set. The comments in this section give the reader the impression that staggering is the most important (or only) factor. I suggest that "for explicit timestepping schemes" is added after "timestep size" on line 1 and that the text on lines 14-15 from "but also" to the semi-colon is removed since I believe this statement is only true for a given choice of horizontal discretization (2nd order fd?) and defined explicit timestepping schemes.**

Agreed. Line 1 has been changed accordingly. The text on lines 14-15 has been changed to

...the C-grid better represents short wave modes, does not support extraneous computational modes (as long as the number of horizontal faces is equal to twice the number of volumes), but typically has a more restrictive timestep with explicit timestepping schemes than the A-grid...

**Page 16 Line 18: In general I think it is a Poisson problem that needs to be solved for the z-grid rather than the more general Helmholtz problem.**

Agreed. "Helmholtz" has been replaced with "Poisson".

**Page 28 Line 14: Could a citation (at least title and authors) be given for this paper if it is under review.**

A citation has been added.

**Typos**

Thank you for catching these. All have been corrected

**Before submitting a revision, I would appreciate it if the lead author of the paper would test the veracity of the links to (or other ways of accessing) the other models included in the code accessibility table. In my experience there are often small issues that need to be resolved to make sure the information is accurate and that the code is really accessible to all.**

As requested, we have ensured the veracity of the links to the model codes.

**Paraphrased comments from anonymous 3rd reviewer. The reviewer commented to the effect that. . . The purpose of the paper is not clear. There is nothing of interest or importance to the modelling community in the paper. The paper is too long, has too many equations and all of it is already published in the literature.**

We disagree strongly with the reviewer regarding the utility of the manuscript. The manuscript itself should be viewed in the light of reviewing existing material from the wealth of (often unavailable) non-hydrostatic modeling technical reports and peer-reviewed publications, rather than introducing substantially new concepts. The structure of this paper provides us with a mechanism to compare the different design decisions that have been made in these models, and potentially as a resource for future students of non-hydrostatic modeling systems to learn about the types of decisions that need to be made. In the view of the authors, the content of our manuscript is simply not easily accessible in the existing literature.

Regarding the length of the paper, we certainly acknowledge that this material could be the basis for a textbook on the subject, but believe that an open-access and peer-reviewed compilation provides broader access to this content than a textbook.

**The material is presented in a highly confusing manner, "for instance, Section 2 is subdivided by model (with 11 sections, 1 per model), but Section 3 only has 8 subsections (why not 11?), and Section 4 only 2 subsections, the second of which only has 4 subsubsection (yet, when you look at Table 4, last column, I only see 3 different methods, so I have no idea where there are 4 sub-subsections for 3 methods); then Section 5 is again subsectioned by model (why this back and forth), but instead of 11 I only see 8 subsections: why are 3 models missing?"**

**[editor's note: GMD papers are allowed to be long and may contain many equations, but because of this, an easy to navigate logical and readable structure is even more important than it is for regular science papers.]**

The issues with the structure have also been highlighted by the second reviewer. In response to his specific criticisms, the revised manuscript now fills in all gaps within sections that are structured around the different options available to each model. We note that each section is meant to highlight one of the potential decisions that need to be made in building a non-hydrostatic model. If the potential options for that decision were limited, it made more sense to the authors to break up the section by the choices available. If essentially all modeling groups pursued a different strategy, then it made more sense to break up each section by the model. We hope the revisions have been sufficient to address this concern.

[revised manuscript text omitted]

---

## Author Response (AR2)

**Comments to the Author:**
**These are technical corrections, but experience has taught me that the editor needs to check any changes to the manuscript that they really would like to see implemented! Hence I have opted for minor revision, as this requires editor review.**

**On the whole, I think the paper is great, and I am looking forward to seeing further DCMIP papers.**

I think I can say, on behalf of all DCMIP contributing authors, that we're also excited to see our results published.

**In relation to the structure, most of the sections deal with the differences between the models, organised by model. Section 3, however, is structured by grid type. It is not possible (for me at least) in every case to easily trace which model uses which grid. Would it be possible, therefore, in the sub-sections of section 2, to cite the subsection of section 3 which describes the grid? Also make sure that the model(s) which use the grid are named in the relevant subsection of section 3.**

Forward references have been added in section 2, and explicit mentions of the models utilizing each model grid have been added to section 3. A few additional forward references have been added among the model descriptions in section 2 for ease of navigation.

**Section 3.1 "This grid is employed by the UK Met Office (Davies et al., 2005; Wood et al., 2014)." This seems a very strange sentence. Many models use a lat-lon grid! One of the references seems to be related to non-hydrostatic calculations at the Met Office, the other one to the regular dynamical core of the UM. Do they have a non-hydrostatic model? Anyway - some qualification/additional explanation is required here.**

Yes, the UK Met Office Unified model supports both a hydrostatic and quasi-non-hydrostatic option. More information can be found at `https://www.metoffice.gov.uk/research/foundation/dynamics/new-dynamics`. This sentence has been changed to read:

[revised manuscript text omitted]

- (a) Z-Lorenz: interface $N$: $w$; level $N$: $\rho\ \theta_v\ \mathbf{u}_h$; $w$; ... level 2: $\rho\ \theta_v\ \mathbf{u}_h$; $w$; level 1: $\rho\ \theta_v\ \mathbf{u}_h$; interface 0: $w$.
- (b) Z-Charney-Phillips: interface $N$: $w\ \theta_v$; level $N$: $\rho\ \mathbf{u}_h$; $w\ \theta_v$; ... level 2: $\rho\ \mathbf{u}_h$; $w\ \theta_v$; level 1: $\rho\ \mathbf{u}_h$; interface 0: $w\ \theta_v$.
- (c) GEM: interface $N$: $\dot{\zeta}$; level $N$: $\mathbf{u}_h\ p$; $w\ T_v\ \dot{\zeta}$ (N-1); ... interface 2: $w\ T_v\ \dot{\zeta}$; level 2: $\mathbf{u}_h\ p$; $w\ T_v\ \dot{\zeta}$ (1); level 1: $\mathbf{u}_h\ p$; $w\ T_v$ (1/4); interface 0: $p\ \dot{\zeta}$.

[revised manuscript text omitted]